# Towards Understanding Modality Interaction in Multimodal Language Models via Partial Information Decomposition

**Wanlong Fang** [1 2]   **Tianle Zhang** [2]   **Wen Tao** [2]   **Alvin Chan** [2 3 4]

## Abstract

Understanding modality interaction in multimodal large language models (MLLMs) is central to reliable deployment. We introduce Partial Information Decomposition (PID) as a decision-level framework that separates unique, redundant, and synergistic contributions of sensory and linguistic inputs, beyond representation alignment and outcome-based evaluation. Across vision–language benchmarks, PID reveals recurring modality-use profiles: reasoning and grounding-oriented tasks tend to exhibit high synergy, whereas expert and knowledge-oriented tasks show stronger language-unique reliance. These profiles generalize across model families and predict sensitivity to modality-level interventions. We further extend PID to tri-modal systems with Sensory PID, treating language as a control variable to decompose video–audio information gain. Applied to omni-modal models, Sensory PID reveals a sensory synergy bottleneck dominated by visual information even on audio–visual fusion tasks. Finally, PID-guided reweighting provides initial evidence for improving multimodal reasoning and grounding performance.

## 1. Introduction

Multimodal large language models (MLLMs) are moving from perception systems toward *decision-making agents* in scientific analysis, medical support, and embodied interaction (Yin et al., 2024; Jin et al., 2025; Wang et al., 2025). By integrating vision, audio, and language within unified generative backbones, these models can ground abstract instructions in rich sensory inputs and produce context-sensitive responses (Liang et al., 2024; Deng et al., 2025; Zhou et al., 2022). Yet current evaluation paradigms still primarily assess *what* models predict, while providing limited insight into *how* modalities are used to reach those predictions (Pan et al., 2024; Hossain et al., 2023; Deitke et al., 2025).

A growing body of work has sought to go beyond outcome accuracy through representation analysis, cross-modal alignment metrics, attention-based inspection, and modality ablation (Yan et al., 2025; Lou et al., 2025; Yoon et al., 2025; Jiang et al., 2025; Li et al., 2025). These approaches reveal how modalities are encoded or how predictions change under perturbations, but they do not provide a decision-level decomposition of modality use. In particular, they cannot distinguish whether a modality contributes information that is *unique*, *redundant* with another modality, or *synergistic*—available only through joint observation. As a result, different forms of multimodal interaction can be conflated under the same accuracy or ablation signal.

We address this gap by formulating multimodal reasoning as a *decision-level information decomposition* problem. Given a model prediction $Y$ and modality-specific sources, Partial Information Decomposition (PID) (Williams & Beer, 2010; Bertschinger et al., 2014) separates the mutual information between inputs and decisions into unique, redundant, and synergistic components. Unlike representation similarity, this analysis is computed over model-induced predictive distributions, allowing us to characterize how information is functionally used in the final decision. For vision–language models, we decompose $I(Y; X_v, X_t)$ into visual-unique, text-unique, redundant, and synergistic terms. For omni-modal models with video, audio, and instructions, we introduce *Sensory PID* (Figure 1), a conditional formulation that treats language as a directive control signal and decomposes the sensory information gain $I(Y; V, A \mid T)$. This avoids the combinatorial expansion of full multi-source PID while preserving the functional distinction between task specification and sensory evidence.

Applying this framework reveals recurring modality-use profiles across MLLMs. Across 20 vision–language models and

---

[1]Artificial Intelligence-X (AI-X) @ NTU, Interdisciplinary Graduate Programme, Nanyang Technological University, Singapore [2]College of Computing and Data Science, Nanyang Technological University, Singapore [3]Lee Kong Chian School of Medicine, Nanyang Technological University, Singapore [4]Centre of AI in Medicine (C-AIM), Nanyang Technological University, Singapore. Correspondence to: Alvin Chan <guoweialvin.chan@ntu.edu.sg>.

*Proceedings of the $43^{rd}$ International Conference on Machine Learning*, Seoul, South Korea. PMLR 306, 2026. Copyright 2026 by the author(s).

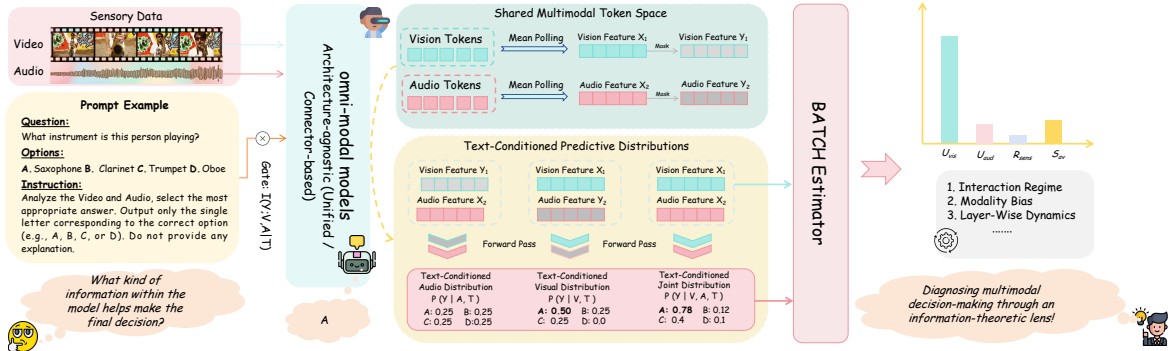

*Figure 1.* **The Sensory PID Framework.** (Left) Omni-modal models are analyzed by treating the text prompt as a gate controlling video–audio integration. (Right) Applying the BATCH estimator in the shared token space decomposes decision-level information into Unique ($U_{vis}$, $U_{aud}$), Redundant ($R_{sens}$), and Synergistic ($S_{av}$) components, revealing multimodal reasoning mechanisms.

6 benchmarks, PID identifies model–benchmark interaction patterns: reasoning and grounding-oriented settings tend to elicit higher cross-modal synergy, whereas expert and domain-knowledge settings exhibit stronger text-unique reliance. These profiles are not imposed by benchmark labels; they emerge from the PID spectrum and vary systematically across model families. Moreover, the decomposed PID terms predict behavior under modality-level interventions: total mutual information tracks overall accuracy, whereas synergy better captures sensitivity to image removal. Extending the analysis to omni-modal models further reveals a sensory synergy bottleneck: on audio–visual fusion tasks, evaluated models remain dominated by modality-unique sensory information, especially visual information, while audiovisual synergy remains comparatively small.

Beyond diagnosis, we also test whether PID-derived signals can guide model improvement. Using BATCH-induced local diagnostic scores, we construct a PID-guided sample reweighting policy that upweights under-synergized samples and downweights language-shortcut samples during lightweight LoRA fine-tuning. This proof-of-concept improves reasoning and grounding performance over uniform fine-tuning and matched reweighting baselines, while shifting post-tuning PID profiles toward higher synergy and lower text-unique reliance. These results suggest that decision-level PID can serve not only as an interpretability tool, but also as a practical signal for identifying when and where multimodal fusion should be strengthened.

We summarize our main contributions as follows:

**Decision-level PID for multimodal decisions.** We formalize modality interaction in MLLMs as a decision-level information decomposition problem, using PID to separate unique, redundant, and synergistic contributions that are not identifiable from representation similarity, accuracy, or ablation alone.

**Sensory PID for omni-modal models.** We introduce a conditional PID formulation for tri-modal systems that treats

language as a directive control signal and decomposes the sensory information gain from video and audio, enabling tractable analysis of omni-modal decision-making.

**Diagnostic, predictive, and actionable findings.** Across vision–language and omni-modal models, PID reveals recurring modality-use profiles, predicts intervention sensitivity, and uncovers a sensory synergy bottleneck. We further propose PID-guided reweighting to improve multimodal inference by encouraging stronger cross-modal synergy.

## 2. Decision-Level PID for Multimodal Language Models

We formalize multimodal interaction as a decision-level information decomposition problem. Rather than analyzing how modalities are *represented* in latent space, we decompose the mutual information between model *decisions* and input modalities into interpretable components. This section presents the framework in three parts: the bi-modal PID formulation (§2.1), its conditional extension to tri-modal systems (§2.2), the estimation pipeline (§2.3).

### 2.1. PID for Vision–Language Prediction

Let $Y$ denote the model's prediction over a finite candidate set $\mathcal{C}$ (*e.g.*, multiple-choice options: *A, B, C, D*), and let $X_v$ and $X_t$ denote the vision and text source variables. We construct an empirical decision distribution by pairing the data distribution with the model's predictive distribution: $P(x_v, x_t, y) = P_{\mathcal{D}}(x_v, x_t)p_\theta(y \mid x_v, x_t)$, with analogous unimodal distributions induced by $p_\theta(y \mid x_v)$ and $p_\theta(y \mid x_t)$ via calibrated embedding masking (§2.3). All mutual information quantities are computed with respect to these model-induced predictive distributions.

Partial Information Decomposition (Williams & Beer, 2010) partitions the joint mutual information into four non-negative components:

$$I(Y; X_v, X_t) = U_{\text{vis}} + U_{\text{txt}} + R_{\text{vl}} + S_{\text{vl}}, \qquad (1)$$

where $U_{\text{vis}}$ and $U_{\text{txt}}$ denote information uniquely attributable to vision and text, $R_{\text{vl}}$ denotes redundant information shared by both modalities, and $S_{\text{vl}}$ denotes synergistic information that arises only from their joint observation.

Following Bertschinger et al. (2014), these components are defined by optimizing over the set $\Delta_P$ of joint distributions $Q(X_v, X_t, Y)$ that preserve the source–target marginals: $\Delta_P = \{Q : Q(x_v, y) = P(x_v, y), Q(x_t, y) = P(x_t, y)\}$. Synergy is the gap between the true joint MI and the minimum achievable under $\Delta_P$:

$$S_{\text{vl}} = I_P(Y; X_v, X_t) - \min_{Q \in \Delta_P} I_Q(Y; X_v, X_t). \quad (2)$$

Let $M_{\min} = \min_{Q \in \Delta_P} I_Q(Y; X_v, X_t)$. The remaining components follow algebraically:

$$U_{\text{txt}} = M_{\min} - I_P(Y; X_v), \quad U_{\text{vis}} = M_{\min} - I_P(Y; X_t),$$
$$R_{\text{vl}} = I_P(Y; X_v) - U_{\text{vis}}.$$

Thus, PID serves as a *decision-level* descriptor of modality use: it is computed from the model's predictive distributions rather than latent representations or ground-truth labels. The relative magnitudes of its components define a *modality-use profile* for each model–benchmark pair, capturing how a specific model uses information on a specific benchmark rather than an intrinsic dataset property.

### 2.2. Sensory PID for Omni-Modal Models

Omni-modal models process a video stream $V$, an audio stream $A$, and a textual instruction $T$ to generate a response $Y$. A full three-source PID decomposition is possible in principle but leads to a combinatorial expansion of partial information components, obscuring the functional role of language in instruction-following settings. We therefore treat $T$ as a conditioning variable and focus on decomposing the *sensory information gain*:

$$I(Y; V, A \mid T) = U_{\text{vis}} + U_{\text{aud}} + R_{\text{sens}} + S_{\text{av}}, \quad (3)$$

where $U_{\text{vis}}$ and $U_{\text{aud}}$ quantify information uniquely attributable to video and audio, $R_{\text{sens}}$ measures redundant sensory evidence, and $S_{\text{av}}$ denotes sensory synergy— information arising exclusively from joint audiovisual integration. This conditional formulation aligns sensory interaction with the directive role of language: $T$ specifies the task, while $V$ and $A$ provide the evidence, enabling a principled separation between task specification and sensory contribution. We estimate this conditional decomposition using the same BATCH pipeline applied to text-conditioned sensory representations.

### 2.3. Estimation Framework

Estimating PID from the high-dimensional, continuous hidden states of modern MLLMs poses two challenges: computing PID on continuous variables and obtaining unimodal

predictive conditionals from a jointly trained model. We address both without retraining the MLLM, using an auxiliary estimator trained only for information estimation.

**Source representations.** We define $X_v$ and $X_t$ as mean-pooled modality token representations after they are mapped or embedded into the shared multimodal token space (*i.e.*, $X_v, X_t \in \mathbb{R}^d$). This aggregation provides a global, sample-aligned modality summary suited for decision-level information estimation. Sensitivity to the pooling strategy is examined in Appendix G.

**PID estimation.** Because $X_v$ and $X_t$ are high-dimensional continuous representations, direct optimization over $\Delta_P$ is intractable. We estimate the PID terms using the BATCH estimator (Liang et al., 2023), which learns a Sinkhorn-normalized coupling $\tilde{Q}$ that preserves the required $X_v$–$Y$ and $X_t$–$Y$ marginals. The PID components are then recovered from $\tilde{Q}$ and the empirical decision distribution $P$ via Eq. (2). We use BATCH as the estimation backend; our methodological contributions lie in applying PID to model-induced MLLM decision distributions.

**Calibrated embedding masking.** To approximate the unimodal conditionals $p_\theta(y \mid x_v)$ and $p_\theta(y \mid x_t)$, we perform calibrated embedding-space masking. To compute $p_\theta(y \mid x_v)$, we replace the *text* modality's projected token embeddings with calibrated Gaussian noise; to compute $p_\theta(y \mid x_t)$, we replace the *vision* embeddings. In each case, the noise for the masked modality $m'$ is drawn as:

$$\tilde{E}_{m'}[\ell] \sim \mathcal{N}(\mu_{m'}, \text{diag}(\sigma_{m'}^2)), \quad \ell = 1, \ldots, L, \quad (4)$$

where $\mu_{m'}$ and $\sigma_{m'}$ are per-dimension statistics of modality $m'$ estimated over the profiling set $\mathcal{D}_{\text{prof}}$. This removes instance-specific information from the masked modality while preserving the distributional statistics expected by the backbone, enabling approximate unimodal inference within the original pipeline.

**Output stabilization.** Three mechanisms ensure reliable estimation. (i) *Valid-option renormalization*: we restrict the predictive distribution to the candidate set $\mathcal{C}$ via selective softmax over the corresponding logits. (ii) *Confidence gating*: before renormalization, if the raw probability mass assigned to the option tokens under the full vocabulary distribution falls below a threshold $\tau = 0.3$, we replace the output with a uniform distribution over $\mathcal{C}$, preventing low-confidence guesses from contributing spurious structure. (iii) *Soft aggregation*: the marginal label distribution is estimated as $\hat{p}(y) = |\mathcal{D}_{\text{prof}}|^{-1} \sum_{j \in \mathcal{D}_{\text{prof}}} p_\theta(y \mid x_v^{(j)}, x_t^{(j)})$, computed once from the profiling set and kept fixed during BATCH optimization, preserving output uncertainty and avoiding discretization artifacts.

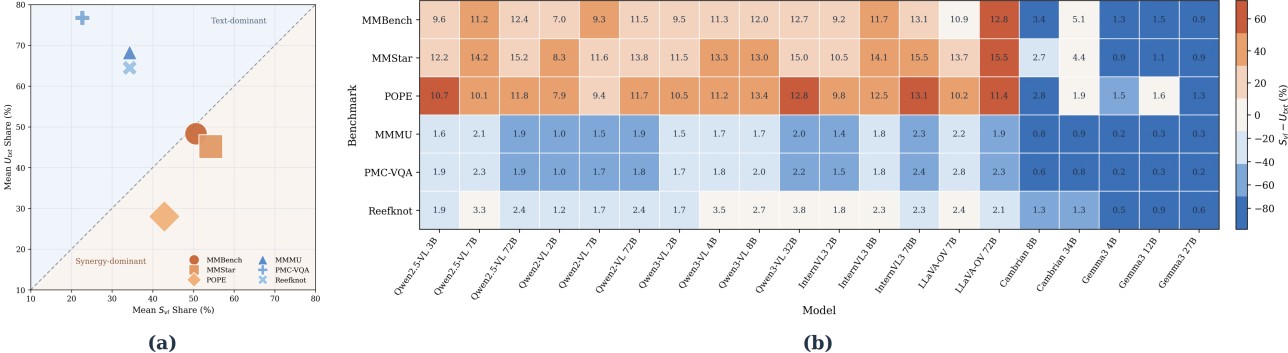

*Figure 2.* PID-derived modality-use analysis across 20 models and 6 benchmarks. **(a)** Benchmark landscape: each marker plots one benchmark's mean $S_{vl}$ share vs. mean $U_{txt}$ share (averaged across all 20 models). The dashed diagonal marks $S_{vl} = U_{txt}$; points with $S_{vl} > U_{txt}$ are synergy-dominant, points with $U_{txt} > S_{vl}$ are text-dominant. **(b)** Model–benchmark interaction profiles. Cell color: profile gap ($S_{vl}$ share $- U_{txt}$ share, pp); red = joint-modality-reliant, blue = language-prior-reliant. Cell values: $\Delta_{\text{vision}} = \text{Acc}(V, T) - \text{Acc}(T)$ (pp); larger values indicate stronger vision dependence.

## 3. Experimental Setup

To test whether PID profiles are robust across architectures and task families, we evaluate a broad set of MLLMs and benchmarks. This design supports three forms of validation: cross-model profiling across architectures and scales, cross-benchmark interaction-profile discovery, and cross-setting extension to omni-modal models. We do not assign modality-use strategy labels or interaction regime labels to models and benchmarks *a priori*; both emerge from the PID analysis in §4.1.

**Models.** We evaluate 20 vision–language models from 7 families, spanning parameter scales from 2B to 78B: Qwen2.5-VL (Bai et al., 2025b), Qwen2-VL (Wang et al., 2024), Qwen3-VL (Bai et al., 2025a), InternVL3 (Zhu et al., 2025), LLaVA-OneVision (Li et al., 2024), Cambrian-1 (Tong et al., 2024), and Gemma3 (Team et al., 2025). These cover early-fusion, late-fusion, and multi-encoder designs. For the omni-modal analysis, we evaluate Qwen2.5-Omni (Xu et al., 2025) and VITA-1.5 (Fu et al., 2025).

**Benchmarks.** We evaluate our framework on six vision–language benchmarks and one omni-modal benchmark. For the vision–language setting, we use MMBench (Liu et al., 2024b) for general LVLM abilities, MMStar (Chen et al., 2024b) for vision-indispensable multimodal reasoning, MMMU (Yue et al., 2024) for expert multidisciplinary understanding, PMC-VQA (Zhang et al., 2023) for medical visual question answering, POPE (Li et al., 2023) for object hallucination and visual grounding, and Reefknot (Zheng et al., 2025) for relation hallucination. For the omni-modal setting, we use MUSIC-AVQA (Li et al., 2022), an audio-visual question-answering benchmark on music videos. We construct three balanced featured subsets: *Audio-Focused*, *Visual-Focused*, and *AV-Fusion*. The AV-Fusion subset is used as the primary testbed for sensory synergy because

its questions are designed to emphasize joint audio–visual reasoning and reduce unimodal shortcuts.

**Implementation and intervention protocol.** All models are evaluated zero-shot with standardized prompts and deterministic greedy decoding. As a behavioral validation, we additionally remove the image input to obtain a text-only baseline and measure the accuracy drop $\Delta_{\text{vision}}$, which is related to PID-derived profiles in §4.1.3. The PID estimation pipeline follows §2.3.

## 4. Modality-Use Profiles: Diagnosis and Prediction

We apply the PID framework from §2 to VL models across 6 benchmarks, progressing from diagnosis to prediction. Full per-model results with all four PID terms and complete intervention data are in Appendices F.

### 4.1. The Landscape of Vision–Language Interaction

#### 4.1.1. BENCHMARK-LEVEL REGIME SEPARATION

We first ask a benchmark-level question: do different evaluation tasks elicit separable modality-use profiles across MLLMs? Figure 2 (a) plots each benchmark's mean $S_{vl}$ share against its mean $U_{txt}$ share, averaged across all 20 models. Two separable clusters emerge without any a priori categorization.

Three benchmarks—MMBench, MMStar, and POPE—fall on the $S_{vl} > U_{txt}$ side of the diagonal, indicating a synergy-dominant model-averaged profile. Three others—MMMU, PMC-VQA, and Reefknot—fall on the $U_{txt} > S_{vl}$ side, with $U_{txt}$ shares exceeding 60%, indicating stronger text-unique reliance. A particularly informative contrast is POPE versus Reefknot: both target hallucination, yet PID separates them into different regimes. POPE exhibits a grounding-

*Table 1.* Spearman $\rho$ between PID quantities and two behavioral measures across 20 models. Each cell reports $\rho(\cdot, \Delta_{\text{vision}})$ / $\rho(\cdot, \text{Acc})$. $I$: total MI; CoI: co-information ($R_{\text{vl}} - S_{\text{vl}}$). $^{\ddagger}p{<}.001$; $^{\dagger}p{<}.01$; $^{*}p{<}.05$; $^{\circ}p{<}.10$. Bold: $|\rho|{\geq}0.68$.

| Regime | Benchmark | $S_{\text{vl}}$ | $U_{\text{txt}}$ | $U_{\text{vis}}$ | $R_{\text{vl}}$ | $I(V,T;Y)$ | CoI |
|---|---|---|---|---|---|---|---|
| Synergy-driven | MMBench | $\mathbf{0.840^{\ddagger}/0.752^{\ddagger}}$ | $-0.582^{\dagger}/0.198$ | $0.068/0.058$ | $0.098/0.088$ | $-0.118/0.522^{*}$ | $\mathbf{-0.822^{\ddagger}/-0.682^{\ddagger}}$ |
| | MMStar | $\mathbf{0.862^{\ddagger}/0.762^{\ddagger}}$ | $-0.548^{*}/0.148$ | $0.042/0.032$ | $0.062/0.052$ | $-0.082/0.478^{*}$ | $\mathbf{-0.850^{\ddagger}/-0.718^{\ddagger}}$ |
| | POPE | $\mathbf{0.798^{\ddagger}/0.718^{\ddagger}}$ | $-0.502^{*}/0.098$ | $0.078/0.068$ | $0.348/0.298$ | $0.052/0.418^{\circ}$ | $\mathbf{-0.722^{\ddagger}}/-0.622^{\dagger}$ |
| Prior-dominant | MMMU | $0.318/0.391$ | $-0.198/0.422^{*}$ | $0.038/0.058$ | $0.048/0.042$ | $-0.052/0.548^{*}$ | $-0.298/-0.242$ |
| | PMC-VQA | $0.382^{\circ}/0.398^{\circ}$ | $-0.178/0.356^{*}$ | $0.052/0.042$ | $0.032/0.028$ | $-0.078/0.593^{\dagger}$ | $-0.348/-0.317$ |
| | Reefknot | $0.418^{\circ}/0.348$ | $-0.278/0.378^{*}$ | $0.068/0.072$ | $0.058/0.048$ | $0.022/0.552^{*}$ | $-0.382^{\circ}/-0.278$ |

sensitive profile, whereas Reefknot exhibits stronger text-unique reliance in the evaluated models. This suggests that surface-level benchmark labels such as "hallucination" can mask distinct modality-use profiles that PID is able to reveal.

> **Finding 1.** Without using benchmark labels, PID reveals two recurring modality-use patterns: MMBench, MMStar, and POPE tend to elicit synergy-dominant profiles, whereas MMMU, PMC-VQA, and Reefknot tend to elicit stronger text-unique reliance across evaluated models.

#### 4.1.2. MODEL–BENCHMARK INTERACTION PROFILES

We next examine how model families interact with these benchmark profiles: whether they tend to integrate visual–textual evidence or rely more on language priors, and whether such modality-use tendencies remain stable across different task regimes.

**Two recurring model-family strategies.** A striking block structure emerges in Figure 2 (b). Most Qwen2.5-VL, Qwen2-VL, Qwen3-VL, InternVL3, and LLaVA-OneVision models exhibit *joint-modality-reliant* profiles: on synergy-driven benchmarks, $S_{\text{vl}}$ exceeds $U_{\text{txt}}$, and the heatmap shows uniformly red cells with large $\Delta_{\text{vision}}$. In contrast, Gemma3 and Cambrian models exhibit more *language-prior-reliant* profiles: $U_{\text{txt}}$ dominates $S_{\text{vl}}$, and vision removal has limited impact ($\Delta_{\text{vision}} < 2$ pp on prior-dominant tasks). These strategies form a *continuum* rather than a binary dichotomy: the heatmap shows that Cambrian-34B occupies a moderate position—with noticeably lighter blue cells on synergy-driven benchmarks compared to the deep blue of Gemma3—consistent with Cambrian's multi-vision-encoder design.

**Scale preserves interaction profiles.** Scaling largely preserves each family's modality-use profile while amplifying its strength. Qwen3-VL maintains the same red–blue pattern from 2B to 32B, with stronger $S_{\text{vl}}$ on synergy-driven benchmarks; Gemma3 remains persistently blue from 4B to 27B. However, off-diagonal blocks show that even joint-modality-reliant models have modest $\Delta_{\text{vision}}$ on prior-dominant benchmarks. Thus, profiles should be interpreted as *model–benchmark interactions*, not intrinsic model or benchmark properties.

> **Finding 2.** The evaluated MLLMs cluster into recurring *joint-modality-reliant* tendencies (Qwen, InternVL3, LLaVA-OV; high $S_{\text{vl}}$, and larger associated $\Delta_{\text{vision}}$) and *language-prior-reliant* tendencies (Gemma3, Cambrian; high $U_{\text{txt}}$, small $\Delta_{\text{vision}}$). Scaling tends to amplify each family's existing profile rather than erase it.

#### 4.1.3. INTERVENTION-PREDICTIVE VALIDITY

Having established diagnostic profiles, we now test whether PID terms carry *predictive* information about model behavior. For each model–benchmark pair, we define $\Delta_{\text{vision}} = \text{Acc}(V, T) - \text{Acc}(T)$; higher values indicate stronger vision dependence. Table 1 reports the Spearman correlation between six information-theoretic quantities and two behavioral measures across all 20 models per benchmark. CoI provides a compact redundancy-versus-synergy contrast and largely mirrors the synergy-dominant regimes because $R_{\text{vl}}$ is small on most MCQ benchmarks.

**$S_{\text{vl}}$ is the most consistently positive predictor on synergy-driven tasks.** On MMBench, MMStar, and POPE, $S_{\text{vl}}$ achieves $\rho(S_{\text{vl}}, \Delta_{\text{vision}}) \geq 0.798$ ($p < 0.001$) and $\rho(S_{\text{vl}}, \text{Acc}) \geq 0.718$ ($p < 0.001$). Models with higher synergy both perform better and lose more accuracy when vision is removed. $U_{\text{txt}}$ predicts $\Delta_{\text{vision}}$ in the opposite direction ($\rho \leq -0.502$, $p < 0.05$): language-reliant models are buffered against vision removal. This opposition is consistent with the PID interpretation that $S_{\text{vl}}$ and $U_{\text{txt}}$ capture distinct modes of modality use.

**$U_{\text{txt}}$ becomes the informative decomposed term on prior-dominant tasks.** On MMMU, PMC-VQA, and Reefknot, $S_{\text{vl}}$ still remains positively related to accuracy but is no longer dominant. Since vision removal has uniformly small effects ($\Delta_{\text{vision}} < 3.5$ pp), no PID term achieves a strong correlation with $\Delta_{\text{vision}}$. This crossover—$S_{\text{vl}}$ dominant on synergy-driven tasks, $U_{\text{txt}}$ on prior-dominant tasks—supports the regime structure identified in §4.1.1.

*Table 2.* **Sensory PID spectrum and performance across Audio-Focused, Visual-Focused, and AV-Fusion subsets.** $U_{\text{vis}}$, $U_{\text{aud}}$, $S_{\text{av}}$, and $R_{\text{sens}}$ denote unimodal visual contribution, unimodal audio contribution, audiovisual synergy, and sensory redundancy, respectively, computed via conditional PID on $I(Y; V, A \mid T)$. The dominant information component for each model–subset pair is highlighted in **bold**. $Acc$ and $Acc1$ report full audio-visual accuracy and visual-only accuracy under audio removing.

| Model | Audio-Focused | | | | | | Visual-Focused | | | | | | AV-Fusion | | | | | | Full Selection | | | | | |
|---|---|---|---|---|---|---|---|---|---|---|---|---|---|---|---|---|---|---|---|---|---|---|---|---|
| | $U_{\text{vis}}$ | $U_{\text{aud}}$ | $S_{\text{av}}$ | $R_{\text{sens}}$ | $Acc$ | $Acc1$ | $U_{\text{vis}}$ | $U_{\text{aud}}$ | $S_{\text{av}}$ | $R_{\text{sens}}$ | $Acc$ | $Acc1$ | $U_{\text{vis}}$ | $U_{\text{aud}}$ | $S_{\text{av}}$ | $R_{\text{sens}}$ | $Acc$ | $Acc1$ | $U_{\text{vis}}$ | $U_{\text{aud}}$ | $S_{\text{av}}$ | $R_{\text{sens}}$ | $Acc$ | $Acc1$ |
| VITA-1.5 7B | 0.54 | **0.59** | 0.20 | 0.005 | 51.6 | 25.4 | **1.34** | 0.25 | 0.15 | 0.005 | 77.5 | 75.8 | **1.35** | 0.60 | 0.22 | 0.004 | 57.8 | 56.2 | **1.09** | 0.55 | 0.22 | 0.003 | 61.4 | 54.2 |
| Qwen2.5-Omni 3B | 0.48 | **0.64** | 0.15 | 0.004 | 48.0 | 22.0 | **1.26** | 0.22 | 0.12 | 0.004 | 70.0 | 68.0 | **1.25** | 0.55 | 0.21 | 0.003 | 52.0 | 50.0 | **1.06** | 0.50 | 0.18 | 0.001 | 55.0 | 48.0 |
| Qwen2.5-Omni 7B | **0.71** | 0.69 | 0.26 | 0.001 | 55.2 | 42.1 | **1.36** | 0.28 | 0.17 | 0.002 | 79.0 | 77.0 | **1.42** | 0.65 | 0.32 | 0.004 | 62.0 | 58.0 | **1.12** | 0.58 | 0.24 | 0.004 | 64.0 | 56.0 |

**Total MI is insufficient.** Total mutual information $I(V, T; Y)$ correlates with accuracy ($\rho = 0.418$–$0.593$) but not with vision dependence ($|\rho| \leq 0.118$ for $\Delta_{\text{vision}}$). Thus, total information tracks overall performance, whereas decomposed PID terms—especially $S_{\text{vl}}$—capture how sensitive a model is to visual intervention. This distinction explains why models with similar accuracy or total MI can differ substantially in modality use.

> **Finding 3.** $S_{\text{vl}}$ predicts accuracy and vision-removal sensitivity on synergy-driven tasks, while $U_{\text{txt}}$ is more informative on prior-dominant tasks. Since total $I(V, T; Y)$ tracks accuracy but not intervention response, the PID *decomposition* captures modality-use signals beyond total information, motivating PID-guided reweighting in §5.

### 4.2. Visual Dominance in Omni-Modal Models

Having established the validity of the PID framework on vision–language models, we extend Sensory PID to omni-modal systems and focus on the AV-Fusion subset of MUSIC-AVQA, a benchmark explicitly designed to require balanced integration of video and audio. While these models incorporate architectural biases toward multi-stream fusion, our analysis reveals a consistent asymmetry in how sensory information contributes to the decision process.

#### 4.2.1. THE SENSORY SYNERGY BOTTLENECK

The AV-Fusion subset is intended to emphasize joint audio–visual reasoning and would therefore be expected to elicit elevated Sensory Synergy ($S_{\text{av}}$). However, Table 2 shows that $S_{\text{av}}$ remains a minor component of total mutual information across all evaluated omni-modal models. Instead, decisions are dominated by unimodal information, revealing a sensory synergy bottleneck even when the task explicitly demands fusion. We also observe systematic visual dominance: unique visual information ($U_{\text{vis}}$) mostly exceeds unique auditory information ($U_{\text{aud}}$) across evaluated subsets. This suggests that current omni-modal models rely more strongly on visual evidence than auditory evidence, limiting their ability to exploit audio–visual complementarity.

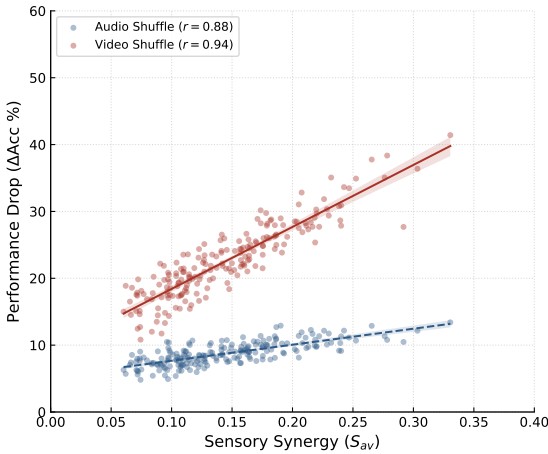

*Figure 3.* **Functional validation via modality shuffling.** Correlation between Sensory Synergy ($S_{\text{av}}$) and accuracy drop ($\Delta\text{Acc}$) on the AV-Fusion subset of MUSIC-AVQA (Qwen2.5-Omni 7B) under audio or video shuffling. Each point represents a question instance; Spearman $\rho$ is reported.

> **Finding 4.** For current omni-modal models, sensory synergy remains minor even on theoretically fusion-dependent tasks, with unimodal information dominating the decision process.

#### 4.2.2. FUNCTIONAL VALIDATION VIA SAMPLE SHUFFLING

To test whether the observed sensory imbalance reflects functional dependence rather than only representational artifacts, we apply sample-level perturbations by independently shuffling video or audio streams across samples. This breaks cross-modal correspondence while preserving modality-specific marginals (Figure 3).

Consistent with the Sensory PID spectrum in Table 2, shuffling the video stream induces substantially larger performance drops than shuffling the audio stream, even on AV-Fusion questions. This asymmetric sensitivity indicates that model predictions depend more strongly on visual information than auditory information under modality-level disruption.

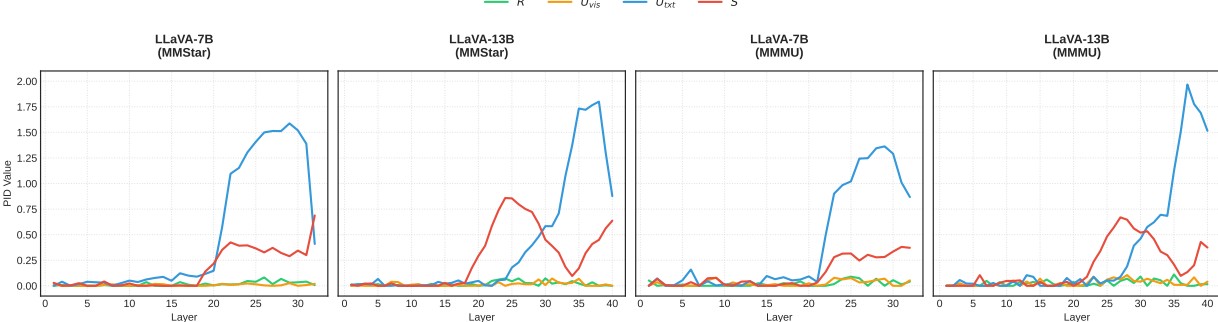

*Figure 4.* **Layer-wise PID in vision–language models.** PID atoms across transformer depth for LLaVA-1.5 (7B/13B) on MMStar and MMMU. Curves show Visual Uniqueness ($U_{\text{vis}}$), Language Uniqueness ($U_{\text{txt}}$), Redundancy ($R_{\text{vl}}$), and Synergy ($S_{\text{vl}}$). Depth is normalized to model layers.

---

> **Finding 5.** Sensory PID predicts functional sensitivity to modality-level disruption: omni-modal models suffer larger performance drops under visual misalignment than auditory misalignment, supporting a visually dominant decision profile.

## 4.3. Dissecting the Mechanism: Layer-Wise Dynamics

The synergy bottleneck and modality imbalance observed above raise a deeper question: is cross-modal fusion globally constrained, or does it emerge only at specific stages of computation? To examine this, we trace the layer-wise evolution of Sensory PID terms, providing a mechanistic view of how sensory information is accumulated and integrated across network depth.

### 4.3.1. THE THREE-PHASE EVOLUTION IN VLMS

We begin by establishing a bi-modal baseline, analyzing standard vision–language models from the LLaVA-1.5 family on MMStar and MMMU. As shown in Figure 4, we observe a consistent three-phase pattern in the layer-wise dynamics of multimodal processing.

**Phase I: Silent Encoding (0–20% depth).** All PID components remain near zero, indicating that the network primarily performs low-level token alignment and feature normalization without extracting task-relevant semantic information.

**Phase II: Unimodal Evidence Accumulation (20–80% depth).** Unique information terms rise steadily as the model constructs modality-specific hypotheses. In particular, $U_{\text{txt}}$ increases rapidly toward a plateau, reflecting a strong reliance on linguistic priors during the decision making. During this stage, Synergy $S_{\text{vl}}$ remains comparatively small, with only a shallow mid-layer rise accompanied by minor non-zero redundancy, suggesting tentative but incomplete cross-modal interaction.

**Phase III: Late-Stage Fusion (80–100% depth).** In the final layers, $S_{\text{vl}}$ rises sharply while $U_{\text{txt}}$ declines, indicating a discrete *fusion event* where previously unimodal evidence is integrated to support the final decision.

**Benchmark Variance:** The magnitude of the Phase 3 spike distinguishes the Synergy-driven and Prior-driven tasks. On MMStar (Synergy-driven), the late synergy spike is sharp and dominant, surpassing $U_{txt}$. On MMMU (Prior-driven), $U_{txt}$ dominance persists until the end, with a significantly weaker synergy response. This layer-wise footprint explains MMMU's insensitivity to visual removal, as decisions are largely fixed by language priors in Phase 2.

> **Finding 6.** In vision–language models, cross-modal integration exhibits a pronounced late-stage profile, with the majority of synergistic information emerging in the final fraction of network depth rather than being a persistent feature of intermediate representations.

### 4.3.2. MECHANISM OF VISUAL HEGEMONY AND THE SENSORY SYNERGY BOTTLENECK

We now trace the same layer-wise dynamics in tri-modal omni-models to uncover the mechanistic origin of the sensory synergy bottleneck. In principle, the expected trajectory mirrors the bi-modal case: early encoding, unimodal extraction, and late integration. However, Figure 5 (a)-(c) reveals a qualitative distortion of the intermediate phase.

**Early Visual Saturation.** Instead of balanced unimodal accumulation, Unique Visual Information ($U_{\text{vis}}$) rises sharply during the unimodal extraction phase and rapidly dominates the information spectrum. This early saturation effectively fixes the decision boundary along a predominantly visual axis, constraining the decision space available for subsequent auditory or synergy evidence which confirms that the model prioritizes visual information in decision-making, using audio information as an auxiliary tool, even for tasks requiring cross-modal fusion.

**Constrained Late Fusion.** As in the bi-modal setting, a late-stage increase in Sensory Synergy ($S_{\text{av}}$) still occurs, con-

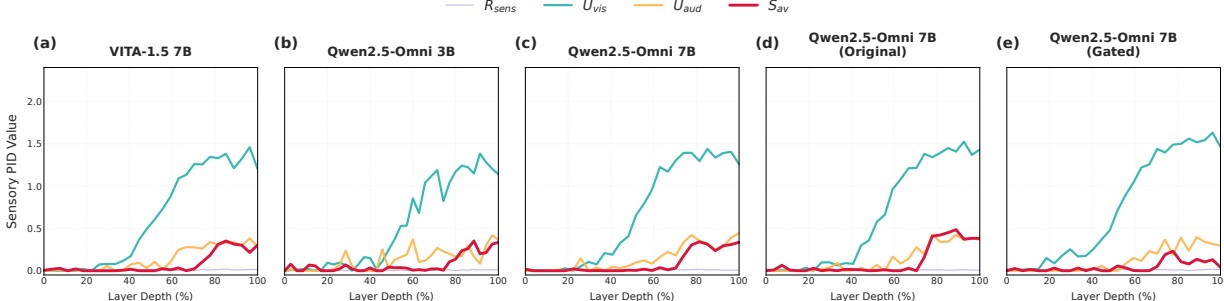

*Figure 5.* **Layer-wise Sensory PID and instruction gating in omni-modal models.** Sensory PID across depth for VITA-1.5 and Qwen2.5-Omni on the AV-Fusion subset of MUSIC-AVQA. Panels (a–c) plot $U_{\text{vis}}$, $U_{\text{aud}}$, $R_{\text{sens}}$, and $S_{\text{av}}$ versus normalized depth. Panels (d–e) show the effect of replacing fusion-demanding instructions with fusion-agnostic paraphrases on $S_{\text{av}}$.

firming that fusion remains a decision-time operation. However, because the decision space has already been shaped by visual dominance, this late synergy has lower magnitude than unique visual contributions.

**Architectural and Scale Effects.** Comparing model families reveals systematic differences in the onset and smoothness of visual dominance. Connector-based architectures (e.g., VITA-1.5) exhibit an earlier and steeper rise in $U_{\text{vis}}$, reflecting stronger visual encoder priors in the intermediate layers. In contrast, unified decoder architectures (e.g., Qwen2.5-Omni) delay this rise but ultimately converge to a similar dominance trajectory at later stages. Increasing model scale reduces variance in the accumulation of visual representations across layers but does not eliminate the persistent gap between $U_{\text{vis}}$ and $U_{\text{aud}}$.

> **Finding 7.** Sensory synergy and unique auditory information emerge later than unique visual information across the network. This ordering suggests a visual-first computation pattern, where intermediate layers are dominated by visual evidence and cross-sensory integration becomes prominent only in later stages.

### 4.3.3. INSTRUCTION GATING OF SENSORY FUSION (TEXT AS A CONTROL SIGNAL)

To determine whether sensory fusion is intrinsically driven by the sensory streams or conditionally modulated by language, we perturb the textual instruction while keeping video and audio inputs fixed. Specifically, we paraphrase or weaken comparative cues that require audiovisual reasoning and re-estimate the layer-wise Sensory Synergy $S_{\text{av}}$.

As an illustrative case study in Figure 5(d)–(e), replacing a fusion-demanding query such as "*Is the violin playing the high pitch?*" with a fusion-agnostic alternative such as "*Which instrument is being played?*" substantially attenuates late-layer sensory synergy, while leaving early- and mid-layer unimodal trajectories largely unchanged.

> **Finding 8.** Language prompts can act as *control signals* that gate sensory fusion, regulating when cross-modal integration is activated rather than merely serving as additional semantic input.

## 5. From Diagnosis to Action: PID-Guided Reweighting

The diagnostic and predictive analyses in §4.1 show that PID profiles capture meaningful structure in multimodal decision-making. We next ask whether this structure can be *exploited* for training to improve multi-modality inference performance. As an initial step, we use BATCH-induced local diagnostic scores (Appendix §A.3) as a sample selection signal for LoRA fine-tuning, without relying on manually defined task categories or sample subtype annotations.

### 5.1. PID-Guided Sample Selection

Standard sample selection methods conflate different sources of difficulty: accuracy-based mining cannot distinguish weak fusion from missing knowledge, while modality-ablation selection detects dependence but not whether it is unique or synergistic. PID helps address this ambiguity by decomposing modality contributions at the decision level. We therefore introduce PID-guided reweighting as a proof-of-concept training strategy.

The BATCH estimator produces not only dataset-level PID atoms but also per-sample local additive contributions $s_i$, $u_{\text{txt},i}$, $u_{\text{vis},i}$, and $r_i$ (see Appendix A.4 for the full derivation). Because these local contributions need not be non-negative, we apply non-negative clipping and define the per-sample information mass $I_i^+ = [s_i]_+ + [u_{\text{vis},i}]_+ + [u_{\text{txt},i}]_+ + [r_i]_+$, where $[\cdot]_+ = \max(\cdot, 0)$. The *synergy ratio* and *shortcut score* are:

$$\text{SR}_i = \frac{[s_i]_+}{I_i^+ + \epsilon}, \qquad \text{SC}_i = \frac{[u_{\text{txt},i}]_+}{I_i^+ + \epsilon}, \qquad (5)$$

where $\epsilon = 10^{-8}$, We stabilize all local scores by averaging over K=50 random batch draws. For each sample $i$, the

*Table 3.* PID-guided reweighting results (Qwen2.5-VL-7B, mean±std over 3 seeds). Post-$S_{\mathrm{vl}}$ and Post-$U_{\mathrm{txt}}$ estimated on a held-out MMStar subset.

| | Reasoning | | Ground. | Prior-Dom. | | PID Diag. | |
|---|---|---|---|---|---|---|---|
| Cond. | MMBench | MMStar | POPE | MMMU | PMC-VQA | Post-$S_{\mathrm{vl}}$ | Post-$U_{\mathrm{txt}}$ |
| A: Base | 88.3 | 60.7 | 86.4 | 54.1 | 53.1 | 1.15 | 0.60 |
| B: LoRA-Uniform | 89.1±.3 | 62.0±.4 | 87.2±.3 | 54.0±.3 | 53.0±.3 | 1.20±.02 | 0.56±.02 |
| **C: LoRA-PID** | **90.2±.2** | **64.3±.3** | **88.5±.2** | 53.5±.4 | 52.7±.3 | **1.36±.03** | **0.46±.02** |
| D: LoRA-Random | 88.8±.5 | 61.5±.6 | 86.9±.4 | 54.1±.4 | 53.2±.4 | 1.16±.04 | 0.58±.03 |
| E: LoRA-Acc | 89.4±.3 | 62.5±.4 | 87.5±.3 | **54.3±.3** | **53.4±.3** | 1.22±.02 | 0.54±.02 |
| F: LoRA-Ablation | 89.6±.3 | 63.0±.4 | 87.8±.3 | 53.8±.3 | 52.9±.3 | 1.24±.02 | 0.52±.02 |

frozen VLM provides three predictive distributions: $p_v^{(i)} = P(Y \mid V_i)$, $p_t^{(i)} = P(Y \mid T_i)$, and $p_{vt}^{(i)} = P(Y \mid V_i, T_i)$. We define a *fusion potential* score capturing how much the joint prediction sharpens over either unimodal prediction:

$$\mathrm{FP}_i = \left[ \min\{H(p_v^{(i)}), \ H(p_t^{(i)})\} - H(p_{vt}^{(i)}) \right]_+, \quad (6)$$

and a *GapScore* identifying samples with high fusion potential but low current synergy usage:

$$\mathrm{GapScore}_i = (1 - \mathrm{SR}_i)(1 - \mathrm{SC}_i) \cdot \mathrm{FP}_i. \quad (7)$$

Shortcut-prone and synergy-gap samples are then selected via top-$K$ ranking: $\mathcal{S}_{\mathrm{short}} = \mathrm{TopK}_{K_s}(\mathrm{SC}_i)$, $\mathcal{S}_{\mathrm{gap}} = \mathrm{TopK}_{K_g}(\mathrm{GapScore}_i)$, $i \notin \mathcal{S}_{\mathrm{short}}$. Training weights are assigned as $w_i = 3.0$ for $i \in \mathcal{S}_{\mathrm{gap}}$ (upweight under-synergized samples), $w_i = 0.5$ for $i \in \mathcal{S}_{\mathrm{short}}$ (downweight language-shortcut samples), and $w_i = 1.0$ otherwise.

### 5.2. Experimental Setup and Main Results

We compare LoRA-PID with zero-shot, standard LoRA, and three matched-count reweighting baselines: random weighting, accuracy-based difficulty mining, and modality-ablation-based selection. Using the BATCH estimator, we profile 3,000 MMBench training samples to derive local PID scores and construct reweighting policies for Qwen2.5-VL-7B, LLaVA-OneVision-7B, and Gemma3-12B. All reweighting methods (C–F) use matched sample counts, with LoRA adapters placed in the last 20% of transformer layers based on our layer-wise fusion analysis. Evaluation spans reasoning-oriented, and prior-dominant benchmarks, with post-tuning PID diagnostics estimated on held-out MMStar. See Appendix H for full algorithm, condition descriptions, LoRA configuration.

**Main results.** Table 3 presents accuracy and post-tuning PID diagnostics across all conditions on Qwen2.5-VL-7B.

Four patterns emerge. (1) PID reweighting yields consistent gains over standard fine-tuning on reasoning and grounding benchmarks (C vs B: +2.3 on MMStar, +1.3 on POPE). (2) The same weight distribution with random assignment does not help (C vs D: +2.8 on MMStar), supporting the interpretation that PID-derived scores provide a useful curation signal. (3) PID-guided selection outperforms accuracy-based difficulty mining (C vs E: +1.8 on

MMStar), suggesting that fusion demand and task difficulty are not interchangeable. (4) PID outperforms ablation-based selection (C vs F: +1.3 on MMStar), though the narrower margin indicates partial overlap between modality sensitivity and synergistic fusion demand. LoRA-PID shows minor decreases on prior-dominant benchmarks (−0.5 on MMMU, −0.3 on PMC-VQA vs. Uniform), consistent with deliberate de-emphasis of language-shortcut samples.

**PID profile shift.** LoRA-PID produces the largest post-tuning synergy increase ($S_{\mathrm{vl}}$: 1.20 → 1.36, +0.16 bits) alongside a corresponding $U_{\mathrm{txt}}$ decrease (0.56 → 0.46), shifting the $S_{\mathrm{vl}}$ share from 67.5% to 73.9%—an increase not replicated by any other condition ($\leq +4.2$ pp for Ablation-RW). Full profile tables are in Appendix H.

> **Finding 9.** PID-derived local scores provide a practical proxy for cross-modal fusion demand. PID-guided reweighting improves cross-modal fusion performance for joint-modality-reliant models, while its limited effect on language-prior-reliant models aligns with the modality-use profiles in §4.1.1.

## 6. Related Work

The shift from vision–language to omni-modal models marks a transition in generative AI (Luo et al., 2025; Chen et al., 2024a; Xu et al., 2025; Guo et al., 2025). Yet evaluation remains outcome-oriented, obscuring modality contributions to decisions. Existing methods, including attention analysis, representation alignment, and ablation, offer partial insights (Wang & Wang, 2025; Basile et al., 2025; Fang et al., 2026). They do not distinguish *redundant* information shared across modalities from *synergistic* information emerging through joint observation. We address this gap with *Sensory PID*, a decision-level framework for quantifying modality interaction. See Appendix B for details.

## 7. Conclusion

We formulate multimodal reasoning as decision-level information decomposition and introduce PID and Sensory PID to disentangle unique, redundant, and synergistic modality contributions beyond alignment and ablation. Across vision–language and omni-modal models, PID reveals recurring modality-use profiles, predicts intervention sensitivity, and exposes a visual-dominant sensory synergy bottleneck. Layer-wise analyses suggest delayed, instruction-gated cross-sensory integration. Finally, PID-guided reweighting provides initial evidence that local diagnostic scores serve as actionable signals for strengthening multimodal fusion. Together, these results position PID as a compact framework for diagnosing, predicting, and guiding modality use in MLLMs. See limitations in Appendix C.

## Acknowledgments

We thank the anonymous reviewers and the program committee for their constructive feedback, which helped strengthen the experimental design and clarify the presentation of this work. We also thank our colleagues for helpful discussions during the development of this project. This research is supported by the Ministry of Education, Singapore, under its Academic Research Fund Tier 2 (MOE-T2EP20125-0002) and Tier 1 (RG22/24), National Research Foundation, Singapore under its National Large Language Models Funding Initiative (AISG Award No: AISG-NMLP-2024-001) and NTU Start Up Grant. Any opinions, findings and conclusions or recommendations expressed in this material are those of the author(s) and do not reflect the views of National Research Foundation, Singapore. The computational work for this article was partially performed on resources of the National Supercomputing Centre (NSCC), Singapore (https://www.nscc.sg).

## Impact Statement

This paper presents work whose goal is to advance the field of machine learning. There are many potential societal consequences of our work, none of which we feel must be specifically highlighted here.

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

# A. Methodology Details and Derivations

This section provides the theoretical underpinnings of PID, details the BATCH estimation procedure, and derives the local diagnostic scores used for reweighting in §5.

## A.1. PID for Continuous Variables

Standard mutual information $I(Y; X_v, X_t)$ measures the total dependence between a target $Y$ and two sources $X_v, X_t$, but cannot decompose this dependence into functionally distinct components. The classical interaction information $I(X_v; X_t; Y) = I(X_v; Y) + I(X_t; Y) - I(X_v, X_t; Y)$ conflates redundancy and synergy into a single signed quantity. PID (Williams & Beer, 2010) resolves this ambiguity by partitioning $I(Y; X_v, X_t)$ into four non-negative terms:

$$I(Y; X_v) = R_{\text{vl}} + U_{\text{vis}}, \tag{8}$$
$$I(Y; X_t) = R_{\text{vl}} + U_{\text{txt}}, \tag{9}$$
$$I(Y; X_v, X_t) = R_{\text{vl}} + U_{\text{vis}} + U_{\text{txt}} + S_{\text{vl}}. \tag{10}$$

Following Bertschinger et al. (2014), we define these terms via optimization over the set $\Delta_P$ of joint distributions $Q(X_v, X_t, Y)$ that preserve the source–target marginals: $Q(x_v, y) = P(x_v, y)$ and $Q(x_t, y) = P(x_t, y)$. Synergy is defined as the gap between the true joint MI and the minimum achievable under $\Delta_P$:

$$S_{\text{vl}} = I_P(Y; X_v, X_t) - \min_{Q \in \Delta_P} I_Q(Y; X_v, X_t). \tag{11}$$

The remaining terms follow algebraically:

$$U_{\text{txt}} = \min_Q I_Q(Y; X_v, X_t) - I_P(Y; X_v), \tag{12}$$

$$U_{\text{vis}} = \min_Q I_Q(Y; X_v, X_t) - I_P(Y; X_t), \tag{13}$$

$$R_{\text{vl}} = I_P(Y; X_v) - U_{\text{vis}} = I_P(Y; X_t) - U_{\text{txt}}. \tag{14}$$

All four terms are non-negative and sum to $I_P(Y; X_v, X_t)$ by construction.

## A.2. Scalable Estimation via BATCH

Solving $\min_{Q \in \Delta_P} I_Q$ directly is intractable for the high-dimensional, continuous representations of modern MLLMs. We employ the BATCH estimator (Liang et al., 2023), which amortizes this optimization using neural networks.

**Neural parameterization.** BATCH learns a parametric distribution $\tilde{Q}_\phi$ via two encoder networks $f_{\phi_1}$ and $f_{\phi_2}$ that map $X_v$ and $X_t$ into a shared latent space. The unnormalized affinity between sample pairs within a mini-batch $\mathcal{B}$ of size $m$ is:

$$A_{ij}^{(y)} = \exp\left(f_{\phi_1}(x_v^{(i)}, y)^\top f_{\phi_2}(x_t^{(j)}, y)\right), \quad i, j \in \{1, \ldots, m\}. \tag{15}$$

**Marginal constraints via Sinkhorn–Knopp.** The PID definition requires $Q \in \Delta_P$. Within a mini-batch, this translates to matching the uniform marginals of the batch samples. BATCH enforces this by applying the Sinkhorn–Knopp algorithm (Cuturi, 2013) to $A^{(y)}$ for each label $y$, iteratively scaling rows and columns until the resulting matrix $\tilde{Q}^{(y)}$ is doubly stochastic.

**Optimization objective.** The estimator minimizes an upper bound on $I_{\tilde{Q}}(Y; X_v, X_t)$ via gradient descent on the encoder parameters:

$$\mathcal{L}(\phi) = \mathbb{E}_{\tilde{Q}}\left[\log \frac{\tilde{Q}(x_t \mid x_v, y)\, \tilde{Q}(x_v \mid y)}{\sum_{y'} \tilde{Q}(x_t \mid x_v, y')\, \tilde{Q}(y' \mid x_v)\, \tilde{Q}(x_v)}\right]. \tag{16}$$

**PID term recovery.** Upon convergence, the four terms are computed by comparing the data distribution $P$ against the optimized $\tilde{Q}$:

$$S_{\text{vl}} = I_P(Y; X_v, X_t) - I_{\tilde{Q}}(Y; X_v, X_t), \tag{17}$$

$$U_{\text{txt}} = I_{\tilde{Q}}(Y; X_v, X_t) - I_P(Y; X_v), \tag{18}$$

$$U_{\text{vis}} = I_{\tilde{Q}}(Y; X_v, X_t) - I_P(Y; X_t), \tag{19}$$

$$R_{\text{vl}} = I_{\tilde{Q}}(Y; X_v, X_t) - U_{\text{txt}} - U_{\text{vis}}. \tag{20}$$

Non-negativity and the additive decomposition are guaranteed by the Sinkhorn constraint.

### A.3. BATCH-Induced Local Diagnostic Scores

Beyond aggregate PID values, the structure of the BATCH computation admits per-sample additive contributions that are useful for downstream applications. The four MI terms underlying the PID—$I_P(Y; X_v)$, $I_P(Y; X_t)$, $I_P(Y; X_v, X_t)$, and $I_{\tilde{Q}}(Y; X_v, X_t)$—are each implemented as sample-level expectations. By retaining per-sample values before averaging, we obtain local contributions:

$$s_i = \text{mi}_P^{(i)}(Y; X_v, X_t) - \text{mi}_{\tilde{Q}}^{(i)}(Y; X_v, X_t), \tag{21}$$

$$u_{\text{txt},i} = \text{mi}_{\tilde{Q}}^{(i)}(Y; X_v, X_t) - \text{mi}_P^{(i)}(Y; X_v), \tag{22}$$

$$u_{\text{vis},i} = \text{mi}_{\tilde{Q}}^{(i)}(Y; X_v, X_t) - \text{mi}_P^{(i)}(Y; X_t), \tag{23}$$

$$r_i = \text{mi}_P^{(i)}(Y; X_v) + \text{mi}_P^{(i)}(Y; X_t) - \text{mi}_{\tilde{Q}}^{(i)}(Y; X_v, X_t), \tag{24}$$

where $\text{mi}_P^{(i)}(Y; X_v) = \sum_y p_\theta(y \mid x_v^{(i)}) \log \frac{p_\theta(y|x_v^{(i)})}{\hat{p}(y)}$ is the local MI contribution from the vision-only predictive distribution, and $\text{mi}_{\tilde{Q}}^{(i)}$ involves the learned Sinkhorn coupling (Appendix A). Within each batch context, the aggregate PID is recovered by averaging these local contributions: $\hat{S} = |\mathcal{B}|^{-1} \sum_i s_i$, and likewise for $\hat{U}_{\text{txt}}$, $\hat{U}_{\text{vis}}$, and $\hat{R}$.

These local scores are *not* axiomatic pointwise PID atoms—a formal theory of pointwise PID remains an open problem (Liang et al., 2023). They are BATCH-induced additive contributions that we use exclusively as *ranking signals*. Because the Sinkhorn coupling depends on batch composition, we stabilize per-sample scores by averaging over $K = 50$ random batch draws (batch size 256).

To enable the reweighting application in §5, we define two normalized diagnostic ratios. Although the aggregate PID atoms are non-negative by construction, their BATCH-induced local additive contributions need not be; we therefore apply non-negative clipping and define the information mass $I_i^+ = [s_i]_+ + [u_{\text{vis},i}]_+ + [u_{\text{txt},i}]_+ + [r_i]_+$, where $[\cdot]_+ = \max(\cdot, 0)$. The *synergy ratio* and *shortcut score* are:

$$\text{SR}_i = \frac{[s_i]_+}{I_i^+ + \epsilon}, \qquad \text{SC}_i = \frac{[u_{\text{txt},i}]_+}{I_i^+ + \epsilon}, \tag{25}$$

where $\epsilon = 10^{-8}$. $\text{SR}_i$ serves as a proxy for the relative synergy usage of sample $i$, while $\text{SC}_i$ captures its relative text-unique reliance. These ratios convert aggregate interaction diagnostics into sample-level ranking signals, enabling the PID-guided sample reweighting policy introduced in §5.

### A.4. BATCH-Induced Local Diagnostic Scores: Full Derivation

Section A.3 introduces per-sample local diagnostic scores used for reweighting in §5. Here we provide the complete derivation and validation.

**Per-sample MI terms.** The four MI quantities underlying the PID—$I_P(Y; X_v)$, $I_P(Y; X_t)$, $I_P(Y; X_v, X_t)$, and $I_{\tilde{Q}}(Y; X_v, X_t)$—are each implemented as sample-level expectations. By retaining per-sample values before averaging, we obtain additive local contributions. The unimodal terms are computed from the predictive distributions obtained via

calibrated embedding masking (§2.3):

$$\mathrm{mi}_P^{(i)}(Y; X_v) = \sum_{y \in \mathcal{C}} p_\theta(y \mid x_v^{(i)}) \log \frac{p_\theta(y \mid x_v^{(i)})}{\hat{p}(y)}, \tag{26}$$

$$\mathrm{mi}_P^{(i)}(Y; X_t) = \sum_{y \in \mathcal{C}} p_\theta(y \mid x_t^{(i)}) \log \frac{p_\theta(y \mid x_t^{(i)})}{\hat{p}(y)}, \tag{27}$$

$$\mathrm{mi}_P^{(i)}(Y; X_v, X_t) = \sum_{y \in \mathcal{C}} p_\theta(y \mid x_v^{(i)}, x_t^{(i)}) \log \frac{p_\theta(y \mid x_v^{(i)}, x_t^{(i)})}{\hat{p}(y)}, \tag{28}$$

where $\hat{p}(y) = |\mathcal{D}_{\mathrm{prof}}|^{-1} \sum_j p_\theta(y \mid x_v^{(j)}, x_t^{(j)})$ is the marginal label distribution estimated once from the profiling set (§2.3).

**Row-anchored coupling MI.** The fourth term involves the Sinkhorn coupling. Fixing the $X_v$ index at sample $i$ (row-anchored) and marginalizing over all $X_t$ samples in the batch:

$$\mathrm{mi}_{\tilde{Q}}^{(i)}(Y; X_v, X_t) = \sum_{j \in \mathcal{B}} \sum_y p_\theta(y \mid x_v^{(i)}) \, \tilde{q}(x_t^{(j)} \mid x_v^{(i)}, y) \left[ \log \frac{p_\theta(y \mid x_v^{(i)}) \, \tilde{q}(x_t^{(j)} \mid x_v^{(i)}, y)}{\hat{p}(y) \, \tilde{q}(x_t^{(j)} \mid x_v^{(i)})} \right], \tag{29}$$

where $\tilde{q}(x_t^{(j)} \mid x_v^{(i)}, y)$ is obtained by normalizing the $i$-th row of the Sinkhorn matrix for label $y$, and $\tilde{q}(x_t^{(j)} \mid x_v^{(i)}) = \sum_{y'} \tilde{q}(x_t^{(j)} \mid x_v^{(i)}, y') \, p_\theta(y' \mid x_v^{(i)})$.

**Local PID scores.** The per-sample contributions (Eqs. 21–24 in the main text) follow:

$$s_i(\mathcal{B}) = \mathrm{mi}_P^{(i)}(Y; X_v, X_t) - \mathrm{mi}_{\tilde{Q}}^{(i)}(Y; X_v, X_t), \tag{30}$$

$$u_{\mathrm{txt},i}(\mathcal{B}) = \mathrm{mi}_{\tilde{Q}}^{(i)}(Y; X_v, X_t) - \mathrm{mi}_P^{(i)}(Y; X_v), \tag{31}$$

$$u_{\mathrm{vis},i}(\mathcal{B}) = \mathrm{mi}_{\tilde{Q}}^{(i)}(Y; X_v, X_t) - \mathrm{mi}_P^{(i)}(Y; X_t). \tag{32}$$

The batch-level PID recovers as $\hat{S} = |\mathcal{B}|^{-1} \sum_i s_i(\mathcal{B})$ and similarly for $\hat{U}_{\mathrm{txt}}, \hat{U}_{\mathrm{vis}}$.

**Row vs. column anchoring.** One could alternatively compute column-anchored contributions (fixing $X_t$, marginalizing over $X_v$). We use row-anchored scores by default and verify consistency: the Spearman correlation between row- and column-anchored $\bar{s}_i$ scores across the 3,000 profiled samples is $\rho = 0.87$ (95% CI $[0.83, 0.91]$), indicating broadly consistent rankings.

**Multi-context averaging.** Because the Sinkhorn coupling depends on batch composition, we stabilize per-sample scores by averaging over $K = 50$ random batch draws (batch size 256):

$$\bar{s}_i = \frac{1}{K} \sum_{k=1}^{K} s_i(\mathcal{B}_k), \quad \bar{u}_{\mathrm{txt},i} = \frac{1}{K} \sum_{k=1}^{K} u_{\mathrm{txt},i}(\mathcal{B}_k). \tag{33}$$

# B. Extended Related Work

**Multimodal Large Language Models and Evaluation.** The progression from vision–language models (VLMs) to omni-modal systems capable of processing interleaved video, audio, and text marks a major shift in generative AI (Fu et al., 2025; Xu et al., 2025). Early systems typically relied on separate modality encoders connected to a language-model backbone, while recent native multimodal architectures increasingly fuse sensory inputs within a shared token space (Zhang et al., 2026a; Dai et al., 2023; Liu et al., 2024a; Luo et al., 2025). Despite strong benchmark performance, evaluation remains largely outcome-oriented, relying on accuracy-based leaderboards (Zheng et al., 2023) or human preference ratings (Zhang et al., 2025a). Such metrics are useful for ranking models, but they provide limited insight into how different modalities contribute to the final decision. Our work shifts the focus from *what* a model predicts to *how* sensory and linguistic information is used at the decision level.

**Interpretability and Modality Interaction Analysis.** Recent work on LLM interpretability has developed a broad set of tools for explaining model behavior beyond final predictions, including attribution, probing, mechanistic analysis, and representation-level inspection (Ma et al., 2026; Zhang et al., 2025b; 2026b). Understanding modality interaction is essential for reliable multimodal deployment. Existing approaches can be broadly grouped into attention analysis, representation

alignment, and modality ablation. Attention-based methods visualize token- or head-level associations (Jain & Wallace, 2019; Chefer et al., 2021; Basile et al., 2025), but attention weights do not necessarily reflect functional contribution. Representation analyses measure geometric or statistical alignment between modality embeddings (Morcos et al., 2018; Nguyen et al., 2020; Radford et al., 2021); however, high alignment does not imply that the aligned information is used in the final prediction. Ablation studies quantify behavioral sensitivity under modality removal or perturbation (Jiao et al., 2024; Tong et al., 2024; Thrush et al., 2022), but they conflate qualitatively different sources of dependence: a performance drop may arise from modality-unique information, redundant evidence, or genuinely synergistic integration. In contrast, our PID framework explicitly separates unique, redundant, and synergistic decision information, enabling a finer-grained characterization of modality use than accuracy, alignment, or ablation alone.

**Information-Theoretic Analysis in Deep Learning.** Information theory provides a principled language for analyzing learned representations and decision processes. The Information Bottleneck principle (Tishby & Zaslavsky, 2015) has been widely used to study compression and prediction in neural networks. Partial Information Decomposition (PID) (Williams & Beer, 2010) extends this perspective by decomposing the information that multiple sources provide about a target into unique, redundant, and synergistic components. Prior PID studies have analyzed interactions in neuroscience, small-scale neural systems, and multimodal datasets, but applying PID to modern MLLMs is challenging because their source variables are high-dimensional and continuous. Recent neural estimators and scalable PID approximations (Faes et al., 2016; Kolchinsky, 2022; Wibral et al., 2017; Bertschinger et al., 2014; Liang et al., 2023) make such analysis increasingly feasible. Recent work has also explored PID-style analysis for vision–language models and diffusion models (Xiu et al., 2026; Dewan et al., 2024). Our work differs from prior information-theoretic analyses in three respects: (i) we formulate PID over model-induced decision distributions in MLLMs, (ii) we introduce Sensory PID to analyze language-conditioned video–audio interaction in omni-modal models, and (iii) we show that PID-derived local diagnostic scores can support sample reweighting, moving the framework beyond post-hoc description toward actionable diagnosis.

## C. Limitations and Future Directions

While PID and Sensory PID provide a principled framework for analyzing multimodal interaction, we acknowledge several limitations of the current study.

**Constrained Decoding Space.** First, our estimation framework operates over a discrete decision space, primarily multiple-choice QA. This setting makes the model's predictive distribution tractable and enables stable PID estimation, but it does not fully capture open-ended generation. Extending PID to free-form responses remains challenging because mutual information must be estimated over high-dimensional, sequential output distributions. Future work will require more efficient estimators for sequence-level or semantic-level decision spaces.

**Approximation of Unimodal Conditionals.** Second, our training-free analysis relies on calibrated embedding masking to approximate unimodal conditionals such as $P(Y \mid V, T)$ or $P(Y \mid A, T)$. Although moment-matched noise helps preserve distributional stability, masking remains a proxy for native unimodal inference. The resulting hidden states may differ from those produced by models explicitly trained or prompted for partial-modality inputs, potentially introducing estimation bias in unique information terms such as $U_{\text{vis}}$ and $U_{\text{aud}}$.

**Statistical Rather than Causal Interpretation.** Third, PID measures statistical dependence in model-induced predictive distributions; it is not a causal calculus. While PID profiles predict sensitivity to modality-level interventions, the PID terms themselves do not identify the internal circuits, attention heads, or neurons responsible for the observed behavior. Thus, high synergy is consistent with functional cross-modal integration, but it does not by itself reveal the mechanistic implementation of that integration.

**Scope of PID-Guided Reweighting.** Finally, our PID-guided reweighting experiment is intended as a proof-of-concept rather than a complete training recipe. We use lightweight LoRA fine-tuning and a limited reweighting policy to test whether local diagnostic scores contain actionable signal. Scaling this idea to larger corpora, stronger adaptation methods, and open-ended instruction tuning remains an important direction for future work.

**Future Directions.**  Building on these limitations, several directions are particularly promising.

- **Extending PID beyond multiple-choice decisions.** Future work could generalize the estimator to continuous or autoregressive output spaces, enabling analysis of chat-style assistants, dense captioning, and long-form multimodal generation.

- **Using PID as a training-time diagnostic.** PID terms could serve as monitoring signals during multimodal training or scaling analysis, helping identify when models rely on language shortcuts, redundant evidence, or genuine cross-modal integration. More speculative extensions could explore PID-inspired auxiliary objectives that encourage fusion-sensitive behavior, while carefully avoiding over-optimization of any single information term.

- **PID-guided benchmark and data construction.** PID analysis can help curate evaluation or training sets that better isolate cross-modal reasoning. For example, samples with high text-unique reliance may reveal language-prior shortcuts, whereas high-synergy samples can support more targeted evaluation of multimodal fusion. This may lead to more informative benchmarks than those defined solely by surface-level task categories.

## D. Experimental Setup Details

### D.1. Prompt Templates

To ensure fair evaluation and reproducible results, we employ a unified prompt structure across all benchmarks. Table 4 details the specific templates used for standard 4-option multiple-choice tasks (e.g., MMBench, MMMU) and binary 2-option tasks (e.g., specific subsets of MUSIC-AVQA or Boolean questions).

*Table 4.* Prompt templates used for zero-shot evaluation. The content within $<\ >$ is replaced by the specific instance data during inference.

| Task Type | Prompt Template |
|---|---|
| **4-Option Questions** (Standard MCQ) | Question: `<Question Body>` 
 Options: 
 (A) `<Option A>` 
 (B) `<Option B>` 
 (C) `<Option C>` 
 (D) `<Option D>` 
 Instruction: Please analyze and select the most appropriate answer. Output only the single letter corresponding to the correct option above (e.g., A, B, C, or D). Do not provide any explanation. |
| **2-Option Questions** (Binary/True-False) | Question: `<Question Body>` 
 Options: 
 (A) `<Option A>` 
 (B) `<Option B>` 
 Instruction: Please analyze and select the most appropriate answer. Output only the single letter corresponding to the correct option above (e.g., A or B). Do not provide any explanation. |

### D.2. Inference Settings

For all evaluations, we use deterministic decoding strategies to minimize variance. Specifically, we set the temperature to $0$ and top-p to $1.0$. The maximum output token length is restricted to ensure the model adheres to the single-letter constraint.

### D.3. Hyperparameters and Training Details

To ensure the full reproducibility of our PID or sensoryji PID estimation results, we provide the detailed hyperparameter configurations used for the BATCH estimator. Following the guidelines established in previous work (Liang et al., 2023), we employed a lightweight Multi-Layer Perceptron (MLP) parameterization to approximate the marginal-matching joint distributions.

We adhered to a fixed configuration across all experiments (Vision-Language and Omni-modal tasks) to ensure consistency. Table 5 summarizes the specific settings. While the estimator is generally robust to hyperparameter variations, we report the exact values used to generate the main results in this paper.

*Table 5.* Hyperparameters and configuration details for the BATCH Estimator.

| Hyperparameter | Value / Configuration |
|---|---|
| *Optimization* | |
| Optimizer | Adam |
| Learning Rate Set | $1 \times 10^{-3}$ and $1 \times 10^{-4}$ |
| Training Epochs Set | 10, 15, 20 |
| Batch Size Set (Training & Inference) | 64/128/256 |
| *Network Architecture* | |
| Encoder Network | 3-layer MLP |
| Hidden Dimension | 32 |
| Activation Function | ReLU |
| Initialization | Standard Xavier |

## E. Dataset Details and Taxonomy

In this section, we provide detailed specifications of the benchmarks used in our analysis. For the vision–language benchmarks, we organize datasets according to the **interaction regimes identified by our PID analysis** in the main text. These categories are analytical summaries of the observed model–benchmark profiles, rather than labels assigned before applying PID. We also describe the subsets constructed for the omni-modal analysis.

### E.1. Vision–Language Benchmarks

**Regime I: Synergy-Dominant Reasoning and Grounding Benchmarks.** These benchmarks exhibit relatively high cross-modal synergy ($S_{\text{vl}}$) in our experiments, suggesting that successful model decisions tend to depend on joint use of visual evidence and textual queries.

- **MMBench** (Liu et al., 2024b): MMBench is a systematically designed benchmark for evaluating LVLMs across approximately 20 fine-grained ability dimensions, ranging from basic object perception and localization to relation reasoning and attribute understanding. Its hierarchical taxonomy and quality-control pipeline make it a broad testbed for general multimodal capability. In our PID analysis, MMBench tends to elicit synergy-dominant profiles across fusion-capable model families, indicating that visual and textual information are often jointly used in the final decision.

- **MMStar** (Chen et al., 2024b): MMStar is a vision-indispensable benchmark designed to reduce multimodal leakage and filter out samples that can be solved from language priors alone. It contains carefully validated multiple-choice questions where visual information is expected to be necessary for correct answering. Consistent with this design goal, our PID results show elevated $S_{\text{vl}}$ on MMStar, making it a representative benchmark for studying cross-modal integration in vision–language models.

- **POPE** (Li et al., 2023): POPE is an object hallucination benchmark that probes whether LVLMs correctly judge the presence or absence of objects in an image through binary yes/no questions. Because the task directly tests object-level grounding, successful predictions often require matching textual object queries with visual evidence rather than relying solely on co-occurrence priors. In our analysis, POPE exhibits a grounding-sensitive, synergy-dominant profile, separating it from other hallucination-oriented benchmarks whose decisions may depend more strongly on language priors.

**Regime II: Text-Unique / Prior-Dominant Knowledge Benchmarks.** These benchmarks exhibit relatively high language-unique information ($U_{\text{txt}}$) in our experiments. Although they contain visual inputs, the evaluated models often rely strongly on textual context, answer priors, or domain knowledge when forming decisions.

- **MMMU (Massive Multi-discipline Multimodal Understanding)** (Yue et al., 2024): MMMU is a large-scale benchmark for college-level multidisciplinary multimodal understanding. It spans multiple disciplines and includes heterogeneous image types such as charts, diagrams, tables, chemical structures, and scientific illustrations. While visual interpretation is important for many questions, the benchmark also requires substantial domain knowledge and reasoning over specialized concepts. In our PID decomposition, MMMU tends to exhibit stronger $U_{\text{txt}}$, suggesting that model decisions are often influenced by language-side domain knowledge and answer priors.

- **PMC-VQA** (Zhang et al., 2023): PMC-VQA is a medical visual question-answering dataset derived from biomedical and clinical imagery, including modalities such as X-rays, MRIs, pathology images, and other medical figures. The questions often require specialized medical knowledge for interpretation and answer selection. Similar to MMMU, our analysis places PMC-VQA in a text-unique / prior-dominant regime, where language-side medical knowledge contributes substantially to the model's predictive distribution.

- **Reefknot** (Zheng et al., 2025): Reefknot evaluates relational hallucination in LVLMs through multiple-choice questions involving spatial or semantic relationships among visual entities. Although it is related to hallucination evaluation, its interaction profile differs from POPE in our PID analysis. Whereas POPE focuses on object-level existence grounding, Reefknot often involves relational patterns that can be partially supported by textual priors or common visual-language associations. Our results show that Reefknot elicits stronger $U_{\text{txt}}$ than POPE, illustrating that benchmarks with similar surface-level labels can induce distinct modality-use profiles.

### E.2. Omni-Modal Benchmark

For the tri-modal analysis $(V, A, T)$, we utilize a specific dataset capable of isolating audio-visual dependencies.

- **MUSIC-AVQA** (Li et al., 2022): A large-scale audio-visual dataset focused on music performance videos. The original dataset contains over 45K QA pairs encompassing spatial-temporal relations and acoustic comparisons.

  **Our Evaluation Split:** To rigorously test the "Sensory Synergy Bottleneck," we do not use the raw random split. Instead, we construct a balanced evaluation set consisting of three distinct subsets (1800 samples each) based on the dataset's metadata:

  1. **Audio-Focused:** Questions solvable primarily via auditory cues (e.g., identifying instruments by sound).
  2. **Visual-Focused:** Questions relying on visual objects or actions (e.g., identifying visible instruments).
  3. **AV-Fusion:** Questions explicitly requiring joint reasoning (e.g., "Is the violin playing the higher pitch?"). This subset serves as our critical testbed for identifying genuine sensory synergy ($S_{av}$).

## F. Full Vision–Language Interaction Tables

Tables 6 and 7 report the complete PID decomposition and vision-removal intervention results for all 20 models across 6 benchmarks. Each cell contains the four PID terms ($S_{\text{vl}}, U_{\text{txt}}, U_{\text{vis}}, R_{\text{vl}}$), full multimodal accuracy (Acc), text-only accuracy ($A_t$), and the vision-removal drop $\Delta = \text{Acc} - A_t$ (pp). Models are grouped by family-level modality-use profile: blue rows denote joint-modality-reliant families and red rows denote language-prior-reliant families, as identified via the PID analysis in §4.1.1. All PID values are in bits; "$\nabla$" denotes <0.01 bits.

These tables extend the representative subset shown in Figure 2 (b) to the full model set, and provide the underlying data for all Spearman correlations reported in Table 1.

**Cross-table patterns.** Several structural regularities are visible across the full model set. (i) $U_{\text{vis}}$ **is uniformly small** ($< 0.03$ bits across all model–benchmark pairs), confirming that vision-unique information plays a negligible role in MCQ decision-making. (ii) $R_{\text{vl}}$ **is format-dependent**: redundancy remains below 0.02 bits on all four-option MCQ benchmarks but rises to 0.18–0.42 bits on POPE (binary Yes/No format), where the reduced option space allows greater overlap between vision-derived and text-derived evidence. (iii) **Family-level profiles are scale-invariant**: within each family, increasing model scale amplifies the dominant PID term without switching the qualitative profile. For example, Qwen3-VL maintains $S_{\text{vl}} > U_{\text{txt}}$ on synergy-driven benchmarks from 2B to 32B, while Gemma3 maintains $U_{\text{txt}} \gg S_{\text{vl}}$ from 4B to 27B. (iv) **Profile–intervention coherence**: joint-modality-reliant models show $\Delta_{\text{vision}} = 7$–15 pp on synergy-driven benchmarks but $< 3.5$ pp on prior-dominant benchmarks; language-prior-reliant models show $\Delta_{\text{vision}} < 5.1$ pp even on synergy-driven benchmarks. This pattern supports the interpretation that PID profiles capture functionally meaningful modality dependence (§4.1.3).

*Table 6.* Full PID decomposition and intervention results on **synergy-driven benchmarks** (20 models × 3 benchmarks). For each model–benchmark pair: $S_{vl}$, $U_{txt}$, $U_{vis}$, $R_{vl}$ are PID terms (bits); Acc = multimodal accuracy (%); $A_t$ = text-only accuracy (%); $\Delta$ = Acc − $A_t$ (pp). Blue = joint-modality-reliant; red = language-prior-reliant. "▽" = <0.01 bits.

| | | MMBench | | | | | | | MMStar | | | | | | | POPE | | | | | | |
|---|---|---|---|---|---|---|---|---|---|---|---|---|---|---|---|---|---|---|---|---|---|---|
| Model | Size | $S_{vl}$ | $U_{txt}$ | $U_{vis}$ | $R_{vl}$ | Acc | $A_t$ | $\Delta$ | $S_{vl}$ | $U_{txt}$ | $U_{vis}$ | $R_{vl}$ | Acc | $A_t$ | $\Delta$ | $S_{vl}$ | $U_{txt}$ | $U_{vis}$ | $R_{vl}$ | Acc | $A_t$ | $\Delta$ |
| Qwen2.5-VL | 3B | .88 | .52 | .01 | .01 | 78.1 | 68.5 | 9.6 | .80 | .45 | .01 | ▽ | 52.4 | 40.2 | 12.2 | .54 | .15 | .02 | .22 | 84.9 | 74.2 | 10.7 |
| Qwen2.5-VL | 7B | 1.10 | .63 | ▽ | .01 | 85.4 | 74.2 | 11.2 | 1.12 | .58 | .02 | ▽ | 60.7 | 46.5 | 14.2 | .57 | .23 | .02 | .35 | 86.4 | 76.3 | 10.1 |
| Qwen2.5-VL | 72B | 1.09 | .83 | ▽ | .02 | 91.9 | 79.5 | 12.4 | 1.12 | .80 | .01 | ▽ | 68.2 | 53.0 | 15.2 | .58 | .20 | .01 | .18 | 89.6 | 77.8 | 11.8 |
| Qwen2-VL | 2B | .85 | .52 | .01 | .01 | 72.5 | 65.5 | 7.0 | .72 | .38 | .01 | ▽ | 44.8 | 36.5 | 8.3 | .52 | .18 | .02 | .27 | 82.4 | 74.5 | 7.9 |
| Qwen2-VL | 7B | 1.12 | .52 | .01 | .01 | 81.3 | 72.0 | 9.3 | .62 | .50 | .01 | ▽ | 56.9 | 45.3 | 11.6 | .45 | .28 | .02 | .30 | 85.2 | 75.8 | 9.4 |
| Qwen2-VL | 72B | .86 | .66 | ▽ | .02 | 88.5 | 77.0 | 11.5 | .98 | .70 | .01 | ▽ | 62.8 | 49.0 | 13.8 | .55 | .22 | .01 | .36 | 88.2 | 76.5 | 11.7 |
| Qwen3-VL | 2B | .72 | .48 | .01 | .01 | 76.5 | 67.0 | 9.5 | .74 | .42 | .01 | ▽ | 48.5 | 37.0 | 11.5 | .46 | .24 | .02 | .30 | 83.5 | 73.0 | 10.5 |
| Qwen3-VL | 4B | .95 | .55 | ▽ | .01 | 82.8 | 71.5 | 11.3 | .98 | .48 | .01 | ▽ | 55.8 | 42.5 | 13.3 | .52 | .22 | .02 | .33 | 86.2 | 75.0 | 11.2 |
| Qwen3-VL | 8B | 1.08 | .58 | .01 | .01 | 86.0 | 74.0 | 12.0 | 1.05 | .52 | .01 | ▽ | 59.0 | 46.0 | 13.0 | .55 | .22 | .02 | .35 | 89.2 | 75.8 | 13.4 |
| Qwen3-VL | 32B | 1.22 | .75 | ▽ | .02 | 90.5 | 77.8 | 12.7 | 1.25 | .70 | .01 | ▽ | 65.5 | 50.5 | 15.0 | .62 | .20 | .01 | .38 | 89.8 | 77.0 | 12.8 |
| InternVL3 | 2B | .55 | .38 | .01 | .01 | 81.7 | 72.5 | 9.2 | .52 | .35 | .01 | ▽ | 50.5 | 40.0 | 10.5 | .48 | .25 | .02 | .28 | 85.3 | 75.5 | 9.8 |
| InternVL3 | 8B | .88 | .42 | ▽ | .01 | 88.0 | 76.3 | 11.7 | .80 | .35 | ▽ | ▽ | 62.3 | 48.2 | 14.1 | .55 | .22 | .01 | .35 | 88.5 | 76.0 | 12.5 |
| InternVL3 | 78B | 1.15 | .70 | ▽ | .02 | 91.1 | 78.0 | 13.1 | 1.38 | .48 | .01 | ▽ | 67.5 | 52.0 | 15.5 | .62 | .18 | .01 | .40 | 89.6 | 76.5 | 13.1 |
| LLaVA-OV | 7B | .97 | .85 | .01 | .01 | 84.7 | 73.8 | 10.9 | .95 | .60 | .01 | ▽ | 58.7 | 45.0 | 13.7 | .52 | .25 | .02 | .32 | 85.7 | 75.5 | 10.2 |
| LLaVA-OV | 72B | 1.47 | .29 | ▽ | .01 | 91.3 | 78.5 | 12.8 | 1.50 | .25 | ▽ | ▽ | 66.8 | 51.3 | 15.5 | .65 | .18 | .01 | .42 | 89.2 | 77.8 | 11.4 |
| Cambrian | 8B | .05 | 1.47 | .01 | .01 | 71.6 | 68.2 | 3.4 | .38 | .55 | .01 | ▽ | 46.5 | 43.8 | 2.7 | .03 | .65 | .02 | .25 | 79.3 | 76.5 | 2.8 |
| Cambrian | 34B | .62 | .82 | .01 | .01 | 79.6 | 74.5 | 5.1 | .55 | .68 | .01 | ▽ | 52.4 | 48.0 | 4.4 | .42 | .38 | .02 | .28 | 86.1 | 84.2 | 1.9 |
| Gemma3 | 4B | .17 | 1.70 | ▽ | .01 | 78.8 | 77.5 | 1.3 | .15 | 1.42 | ▽ | ▽ | 49.4 | 48.5 | 0.9 | .26 | .74 | .01 | .30 | 82.0 | 80.5 | 1.5 |
| Gemma3 | 12B | .19 | 1.75 | ▽ | .02 | 82.5 | 81.0 | 1.5 | .18 | 1.68 | ▽ | ▽ | 54.8 | 53.7 | 1.1 | .53 | .49 | .01 | .35 | 82.1 | 80.5 | 1.6 |
| Gemma3 | 27B | .15 | 1.78 | ▽ | .01 | 84.1 | 83.2 | 0.9 | .14 | 1.72 | ▽ | ▽ | 56.7 | 55.8 | 0.9 | .11 | .78 | ▽ | .22 | 83.8 | 82.5 | 1.3 |

*Table 7.* Full PID decomposition and intervention results on **prior-dominant benchmarks** (20 models × 3 benchmarks). Format and conventions follow Table 6.

| | | MMMU | | | | | | | PMC-VQA | | | | | | | Reefknot | | | | | | |
|---|---|---|---|---|---|---|---|---|---|---|---|---|---|---|---|---|---|---|---|---|---|---|
| Model | Size | $S_{vl}$ | $U_{txt}$ | $U_{vis}$ | $R_{vl}$ | Acc | $A_t$ | $\Delta$ | $S_{vl}$ | $U_{txt}$ | $U_{vis}$ | $R_{vl}$ | Acc | $A_t$ | $\Delta$ | $S_{vl}$ | $U_{txt}$ | $U_{vis}$ | $R_{vl}$ | Acc | $A_t$ | $\Delta$ |
| Qwen2.5-VL | 3B | .28 | .60 | .01 | ▽ | 41.4 | 39.8 | 1.6 | .25 | .55 | .01 | ▽ | 41.4 | 39.5 | 1.9 | .45 | .65 | .01 | .01 | 52.4 | 50.5 | 1.9 |
| Qwen2.5-VL | 7B | .35 | .72 | ▽ | ▽ | 54.1 | 52.0 | 2.1 | .32 | .65 | ▽ | ▽ | 53.1 | 50.8 | 2.3 | .62 | .59 | .01 | .01 | 65.4 | 62.1 | 3.3 |
| Qwen2.5-VL | 72B | .28 | .88 | ▽ | ▽ | 62.4 | 60.5 | 1.9 | .26 | .89 | ▽ | ▽ | 58.4 | 56.5 | 1.9 | .52 | .82 | ▽ | .01 | 74.2 | 71.8 | 2.4 |
| Qwen2-VL | 2B | .15 | .48 | .01 | ▽ | 34.0 | 33.0 | 1.0 | .12 | .45 | .01 | ▽ | 34.5 | 33.5 | 1.0 | .30 | .55 | .01 | .01 | 42.0 | 40.8 | 1.2 |
| Qwen2-VL | 7B | .22 | .65 | .01 | ▽ | 44.0 | 42.5 | 1.5 | .20 | .60 | .01 | ▽ | 44.0 | 42.3 | 1.7 | .38 | .62 | .01 | .01 | 55.2 | 53.5 | 1.7 |
| Qwen2-VL | 72B | .28 | .78 | ▽ | ▽ | 58.4 | 56.5 | 1.9 | .25 | .72 | ▽ | ▽ | 52.8 | 51.0 | 1.8 | .48 | .72 | ▽ | .01 | 68.4 | 66.0 | 2.4 |
| Qwen3-VL | 2B | .25 | .55 | .01 | ▽ | 38.5 | 37.0 | 1.5 | .22 | .50 | .01 | ▽ | 38.2 | 36.5 | 1.7 | .40 | .60 | .01 | .01 | 48.5 | 46.8 | 1.7 |
| Qwen3-VL | 4B | .30 | .62 | ▽ | ▽ | 45.5 | 43.8 | 1.7 | .28 | .58 | ▽ | ▽ | 45.0 | 43.2 | 1.8 | .50 | .62 | .01 | .01 | 55.5 | 52.0 | 3.5 |
| Qwen3-VL | 8B | .32 | .68 | ▽ | ▽ | 50.5 | 48.8 | 1.7 | .30 | .62 | ▽ | ▽ | 49.5 | 47.5 | 2.0 | .55 | .65 | .01 | .01 | 63.9 | 61.2 | 2.7 |
| Qwen3-VL | 32B | .32 | .82 | ▽ | ▽ | 60.5 | 58.5 | 2.0 | .30 | .78 | ▽ | ▽ | 57.2 | 55.0 | 2.2 | .58 | .75 | ▽ | .01 | 72.0 | 68.2 | 3.8 |
| InternVL3 | 2B | .22 | .55 | .01 | ▽ | 43.4 | 42.0 | 1.4 | .20 | .52 | .01 | ▽ | 42.0 | 40.5 | 1.5 | .42 | .50 | .01 | .01 | 52.0 | 50.2 | 1.8 |
| InternVL3 | 8B | .26 | .58 | ▽ | ▽ | 57.4 | 55.6 | 1.8 | .25 | .55 | ▽ | ▽ | 52.0 | 50.2 | 1.8 | .50 | .55 | ▽ | .01 | 64.1 | 61.8 | 2.3 |
| InternVL3 | 78B | .28 | .82 | ▽ | ▽ | 68.8 | 66.5 | 2.3 | .26 | .78 | ▽ | ▽ | 57.4 | 55.0 | 2.4 | .48 | .72 | ▽ | .01 | 68.8 | 66.5 | 2.3 |
| LLaVA-OV | 7B | .32 | .45 | .01 | ▽ | 53.4 | 51.2 | 2.2 | .30 | .42 | .01 | ▽ | 54.8 | 52.0 | 2.8 | .55 | .58 | .01 | .01 | 56.9 | 54.5 | 2.4 |
| LLaVA-OV | 72B | .30 | .87 | ▽ | ▽ | 59.4 | 57.5 | 1.9 | .28 | .85 | ▽ | ▽ | 56.8 | 54.5 | 2.3 | .48 | .82 | ▽ | .01 | 70.1 | 68.0 | 2.1 |
| Cambrian | 8B | .08 | .58 | .01 | ▽ | 43.0 | 42.2 | 0.8 | .06 | .55 | .01 | ▽ | 39.6 | 39.0 | 0.6 | .12 | .65 | .01 | ▽ | 46.5 | 45.2 | 1.3 |
| Cambrian | 34B | .10 | .72 | .01 | ▽ | 49.4 | 48.5 | 0.9 | .08 | .70 | .01 | ▽ | 46.4 | 45.6 | 0.8 | .18 | .82 | .01 | ▽ | 52.4 | 51.1 | 1.3 |
| Gemma3 | 4B | .02 | 1.75 | ▽ | ▽ | 47.2 | 47.0 | 0.2 | .02 | 1.86 | ▽ | ▽ | 43.4 | 43.2 | 0.2 | .12 | 1.45 | ▽ | ▽ | 40.5 | 40.0 | 0.5 |
| Gemma3 | 12B | .03 | 1.85 | ▽ | ▽ | 54.6 | 54.3 | 0.3 | .03 | 1.95 | ▽ | ▽ | 51.4 | 51.1 | 0.3 | .16 | 1.55 | ▽ | ▽ | 46.7 | 45.8 | 0.9 |
| Gemma3 | 27B | .02 | 1.90 | ▽ | ▽ | 57.4 | 57.1 | 0.3 | .02 | 2.00 | ▽ | ▽ | 53.4 | 53.2 | 0.2 | .14 | 1.60 | ▽ | ▽ | 49.8 | 49.2 | 0.6 |

# G. Ablation Studies and Sensitivity Analysis

To validate the robustness of our Sensory PID methodology, we examine the sensitivity of the estimated information components to two critical implementation choices: (i) the strategy for feature summarization (Mean Pooling vs. Last Hidden State vs. Max Pooling), and (ii) the confidence gating threshold $\tau \in \{0.3, 0.4, 0.5\}$.

Table 8 presents the results on MMBench for three representative models. We report the values for Synergy and Text Uniqueness ($S_{vl}/U_{txt}$). The results demonstrate that our estimates are highly stable across different feature aggregation methods and confidence thresholds, with variations remaining negligible (typically $< 0.01$ bits). This confirms that the observed interaction regimes are intrinsic properties of the models rather than artifacts of the estimation hyperparameters.

*Table 8.* Ablation study of PID components ($S_{vl}/U_{txt}$) on MMBench. We compare sensitivity to different feature summarization strategies (left) and confidence thresholds $\tau$ (right). The estimates remain consistent across settings.

| Model | Representation Feature | | | Confidence Threshold ($\tau$) | | |
|---|---|---|---|---|---|---|
| | Mean Pooling | Last Hidden | Max Pooling | 0.3 | 0.4 | 0.5 |
| Qwen2.5-VL-7B | 1.1012/0.6338 | 1.1024/0.6359 | 1.1018/0.6333 | 1.1012/0.6338 | 1.1017/0.6336 | 1.1018/0.6381 |
| Gemma3-12B | 0.1912/1.7513 | 0.1908/1.7516 | 0.1926/1.7521 | 0.1912/1.7513 | 0.1912/1.7548 | 0.1912/1.7598 |
| LLaVA-ov-7B | 0.9736/0.6321 | 0.9737/0.6326 | 0.9735/0.6311 | 0.9736/0.6321 | 0.9736/0.6324 | 0.9742/0.6344 |

# H. PID-Guided Reweighting: Full Details

This section provides complete implementation details, validation experiments, ablation studies, and supplementary results for the PID-guided reweighting in §5. The main text presents the core results (Tables 3); here we report all supporting evidence for reproducibility and interpretation.

## H.1. Algorithm and Experimental Conditions

Algorithm 1 summarizes the PID-guided sample reweighting sample selection procedure described in §5.1. Table 9 provides full descriptions of all six experimental conditions.

---

**Algorithm 1** PID-Guided Sample Selection

---

**Require:** Per-sample scores $\{SC_i, GapScore_i\}_{i=1}^{N}$; selection counts $K_s, K_g$
1: $\mathcal{S}_{\text{short}} \leftarrow \text{TopK}_{K_s}(SC_i)$ {Priority 1: language-shortcut}
2: $\mathcal{R} \leftarrow \{1, \ldots, N\} \setminus \mathcal{S}_{\text{short}}$
3: $\mathcal{S}_{\text{gap}} \leftarrow \text{TopK}_{K_g}(GapScore_i \mid i \in \mathcal{R})$ {Priority 2: under-synergized}
4: $\mathcal{S}_{\text{normal}} \leftarrow \mathcal{R} \setminus \mathcal{S}_{\text{gap}}$
5: Assign weights: $w_i = 3.0$ if $i \in \mathcal{S}_{\text{gap}}$; $w_i = 0.5$ if $i \in \mathcal{S}_{\text{short}}$; $w_i = 1.0$ otherwise

---

*Table 9.* Six experimental conditions. Conditions C–F use matched counts: $K_g = 389$ samples at $3.0\times$ and $K_s = 746$ at $0.5\times$.

| ID | Condition | Description |
|---|---|---|
| A | Base | Frozen pretrained model. Zero-shot reference. |
| B | LoRA-Uniform | Standard LoRA, all samples $w = 1.0$. Isolates fine-tuning gain. |
| C | **LoRA-PID** | Top-$K_g$ by GapScore (Eq. 7) $\uparrow$; top-$K_s$ by SC $\downarrow$. |
| D | LoRA-Random | Same weight distribution as C, randomly assigned. Controls for non-uniformity. |
| E | LoRA-Acc | Top-$K_g$ by difficulty (incorrect samples ranked first; ties broken by $H(p_{vt}^{(i)})$); top-$K_s$ easiest (correct with lowest entropy). |
| F | LoRA-Ablation | Top-$K_g$ by AblScore$_i = H(p_t^{(i)}) - H(p_{vt}^{(i)})$ (joint-over-text entropy reduction); top-$K_s$ by lowest AblScore. Tests whether simple entropy reduction from adding vision suffices without distinguishing unique vs. synergistic contributions. |

## H.2. LoRA Configuration and Data Controls

**LoRA configuration.** All conditions (B–F) use identical LoRA settings: rank $r = 16$, scaling factor $\alpha = 32$, dropout 0.05, learning rate $2 \times 10^{-5}$ with cosine schedule and 5% warmup, effective batch size 16. LoRA adapters target in the last 20% of each model's transformer layers. This targeting is motivated by the layer-wise analysis (§4.3.1), which shows cross-modal synergy emerging in the final layers. Per-model layer ranges: Qwen2.5-VL-7B (layers 23–28 of 28), LLaVA-OV-7B (layers 26–32 of 32), Gemma3-12B (layers 25–30 of 30). Three random seeds $\{42, 123, 456\}$ are used for all conditions; Condition D uses 3 random weight assignments per seed (9 runs total).

**No-leakage guarantee.** No evaluation samples from MMBench-test, MMStar, POPE, MMMU, or PMC-VQA are used for PID profiling, hyperparameter selection, or LoRA training.

**Evaluation protocol.** Post-tuning PID diagnostics (Post-$S_{vl}$, Post-$U_{txt}$) are estimated on a held-out MMStar eval subset via the BATCH estimator using the same hyperparameters as the main analysis . Base PID values are consistent with the main paper. All accuracy results are reported as mean $\pm$ std over 3 seeds. Confidence intervals are computed via seed-stratified paired bootstrap (10,000 resamples).

## H.3. Effective Training Distribution

**Reweighting scheme.** Table 10 reports the effective training distribution.

*Table 10.* Reweighting scheme and effective training distribution.

| Category | Count | Raw % | Weight | Eff. Wt. | Eff. % |
|---|---|---|---|---|---|
| Synergy-Gap | 389 | 13% | $3.0\times$ | 1,167 | 34.0% |
| Normal | 1,865 | 62% | $1.0\times$ | 1,865 | 54.9% |
| Shortcut | 746 | 25% | $0.5\times$ | 373 | 11.1% |
| **Total** | 3,000 | 100% | — | 3,405 | 100% |

## H.4. Supplementary Results

**Pairwise comparisons with confidence intervals.** Table 11 reports all pairwise deltas on MMStar with 95% paired bootstrap CIs (seed-stratified; 10,000 resamples).

*Table 11.* Pairwise comparisons on MMStar with 95% paired bootstrap CIs.

| Comparison | $\Delta$MMStar | 95% CI | $\Delta$Post-$S_{\text{vl}}$ | Interpretation |
|---|---|---|---|---|
| C vs B | +2.3 | [1.4, 3.2] | +0.16 | PID > standard fine-tuning |
| C vs D | +2.8 | [1.6, 4.0] | +0.20 | PID signal > random assignment |
| C vs E | +1.8 | [0.7, 2.9] | +0.14 | PID > difficulty-based |
| C vs F | +1.3 | [0.2, 2.4] | +0.12 | PID > ablation-based |
| C vs A | +3.6 | [2.7, 4.5] | +0.21 | Overall PID-guided gain |

**Per-category accuracy.** Table 12 reports per-category results on the MMBench test set, showing regime-selective gains.

*Table 12.* Per-category accuracy on MMBench test (Qwen2.5-VL-7B). $\Delta_{\text{PID}}$ = LoRA-PID − LoRA-Uniform.

| Subtype | Regime | LoRA-Base | LoRA-Unif. | LoRA-PID | LoRA-Acc | LoRA-Abl. | $\Delta_{\text{PID}}$ |
|---|---|---|---|---|---|---|---|
| Spatial Reasoning | Reasoning | 82.1 | 84.0 | **87.5** | 84.8 | 85.5 | +3.5 |
| Attr. Comparison | Reasoning | 84.5 | 86.2 | **89.0** | 86.8 | 87.3 | +2.8 |
| Action Recognition | Reasoning | 85.8 | 87.0 | **89.5** | 87.5 | 88.0 | +2.5 |
| Object Localization | Percep. | 91.2 | 92.0 | **93.1** | 92.3 | 92.6 | +1.1 |
| Scene Classification | Percep. | 90.5 | 91.0 | 91.5 | 91.2 | 91.3 | +0.5 |
| OCR / Text Recog. | Knowl. | 89.0 | 89.5 | 89.2 | **89.8** | 89.4 | −0.3 |
| Commonsense | Knowl. | 88.3 | 88.8 | 88.5 | **89.2** | 88.7 | −0.3 |

**POPE subset breakdown.** Table 13 shows PID-guided gains are largest on POPE-Adversarial (+1.9 over Uniform), consistent with shortcut reduction mitigating co-occurrence-based hallucination priors.

*Table 13.* POPE subset results (Qwen2.5-VL-7B).

| Cond. | Random | Popular | Adversarial | Avg |
|---|---|---|---|---|
| Base | 88.2 | 86.5 | 84.5 | 86.4 |
| LoRA-Uniform | 88.8 | 87.2 | 85.6 | 87.2 |
| **LoRA-PID** | **89.6** | **88.3** | **87.5** | **88.5** |
| LoRA-Acc-RW | 89.0 | 87.5 | 86.0 | 87.5 |
| LoRA-Abl-RW | 89.2 | 87.7 | 86.5 | 87.8 |

**PID profile shift.** Table 14 reports the full post-tuning PID profile on the held-out MMStar eval subset.

*Table 14.* Post-tuning PID profile (MMStar eval subset, Qwen2.5-VL-7B).

| Condition | $S_{\text{vl}}$ | $U_{\text{txt}}$ | $U_{\text{vis}}$ | $R_{\text{vl}}$ | $S_{\text{vl}}$ **Share** | $\Delta$**Acc** |
|---|---|---|---|---|---|---|
| Base | 1.15 | 0.60 | 0.015 | 0.002 | 65.1% | 7.7 |
| LoRA-Uniform | 1.20 | 0.56 | 0.016 | 0.002 | 67.5% | 8.2 |
| **LoRA-PID** | **1.36** | **0.46** | 0.018 | 0.003 | **73.9%** | **10.8** |
| LoRA-Acc-RW | 1.22 | 0.54 | 0.016 | 0.002 | 68.3% | 8.6 |
| LoRA-Abl-RW | 1.24 | 0.52 | 0.017 | 0.002 | 69.3% | 9.0 |

## H.5. Cross-Model Full Results

Table 15 provides full results for LLaVA-OneVision-7B and Gemma3-12B.

Table 15. Full cross-model results (mean $\pm$ std over 3 seeds).

| Model | Cond. | MMBench | MMStar | MMMU | P-$S_{vl}$ | P-$U_{txt}$ |
|---|---|---|---|---|---|---|
| LLaVA-OV-7B | LoRA-Base | 84.7 | 58.7 | 53.4 | 0.95 | 0.61 |
| | LoRA-Uniform | 85.5$\pm$.3 | 60.0$\pm$.4 | 53.1$\pm$.3 | 1.00$\pm$.02 | 0.58$\pm$.02 |
| | **LoRA-PID** | **86.3**$\pm$.3 | **61.5**$\pm$.4 | 52.8$\pm$.4 | **1.10**$\pm$.03 | **0.50**$\pm$.02 |
| Gemma3-12B | LoRA-Base | 82.5 | 54.8 | 54.6 | 0.18 | 1.68 |
| | LoRA-Uniform | 83.1$\pm$.4 | 56.2$\pm$.5 | 59.3$\pm$.3 | 0.20$\pm$.02 | 1.65$\pm$.03 |
| | **LoRA-PID** | **83.3**$\pm$.4 | 54.3$\pm$.5 | 54.1$\pm$.4 | 0.21$\pm$.02 | 1.63$\pm$.03 |

The Gemma3-12B negative case ($+0.3$, within seed variation) suggests PID-guided reweighting amplifies existing fusion capacity rather than creating it, reinforcing the diagnostic utility of the modality-use profiles (§4.1.1).

## H.6. Ablation Studies

All ablations are conducted on Qwen2.5-VL-7B.

**Upweight factor sensitivity.** Table 16 varies the upweight factor applied to synergy-gap samples ($K_g = 389$, $K_s = 750$, downweight $= 0.5\times$). Performance peaks at $3\times$ and plateaus at $5\times$, while MMMU shows monotonic decrease.

Table 16. Upweight factor sensitivity.

| Upweight | MMStar | Post-$S_{vl}$ | MMMU |
|---|---|---|---|
| $1\times$ (= Uniform) | 62.0 | 1.20 | 54.0 |
| $2\times$ | 63.2 | 1.28 | 53.8 |
| **$3\times$** | **64.3** | **1.36** | 53.5 |
| $5\times$ | 64.0 | 1.40 | 52.8 |

**Selection count sensitivity.** Table 17 varies the shortcut selection count $K_s$. Setting $K_s$ too aggressively (40%) hurts knowledge benchmarks more than it helps reasoning; being too conservative (10%) limits impact.

Table 17. Selection count sensitivity ($K_s$ as % of $N$, upweight $= 3\times$).

| $K_s$ (% of $N$) | Downweighted | MMStar | MMMU |
|---|---|---|---|
| 40% | $\sim$1,200 | 63.8 | 52.9 |
| **25%** | **$\sim$750** | **64.3** | 53.5 |
| 10% | $\sim$300 | 63.0 | 53.9 |

## H.7. Per-Seed Results

Table 18 confirms that LoRA-PID achieves the highest MMStar accuracy across all three seeds individually, with a worst-case advantage of $+1.3$ over LoRA-Ablation (Seed 123).

*Table 18.* Per-seed results on MMStar and MMMU (Qwen2.5-VL-7B). Condition D: mean over 3 random assignments per seed.

| | MMStar | | | MMMU | | |
|---|---|---|---|---|---|---|
| **Condition** | Seed 42 | Seed 123 | Seed 456 | Seed 42 | Seed 123 | Seed 456 |
| B LoRA-Uniform | 62.4 | 61.7 | 61.9 | 54.2 | 53.8 | 54.0 |
| C LoRA-PID | **64.6** | **64.0** | **64.3** | 53.2 | 53.9 | 53.4 |
| D LoRA-Random | 61.8 | 61.2 | 61.5 | 54.3 | 53.7 | 54.3 |
| E LoRA-Acc | 62.9 | 62.2 | 62.4 | 54.5 | 54.0 | 54.4 |
| F LoRA-Ablation | 63.4 | 62.7 | 62.9 | 54.0 | 53.5 | 53.9 |

