# OpenReview forum: "Towards Understanding Modality Interaction in Multimodal Language Models via Partial Information Decomposition"
_ICML.cc/2026/Conference — ICML 2026 regular_

### Official Review · Reviewer_DHqu · 2026-03-02

**Soundness:** 2
**Presentation:** 3
**Significance:** 3
**Originality:** 2
**Overall Recommendation:** 4
**Confidence:** 4

**Summary:**

The authors emphasise the increasing need for modality interaction analysis in MLLMs to identify the effect of modality representations on particular outcomes, particularly how the information is utilised under the partial information decomposition (PID). Moreover, as MLLMs operate under specific textual instructions, the authors extend PID to sensory PID that explores information gain conditioned on the textual information. The authors find that across various model families and sizes, reasoning tasks involve high synergistic information while knowledge-intensive tasks involve high language-unique information. In text-video-audio setups, they observe that visual information dominates audio information, hence prevent the emergence of high-synergy regime.

**Compliance With Llm Reviewing Policy:**

Affirmed.

**Final Justification:**

I thank the authors for the discussion throughout the discussion process; it helped clarify understanding certain points that I missed earlier. Although the authors addressed some of my questions regarding the framing of the paper, I think the paper can benefit from a general re-formulation of the limitations of their work in addition to a wider exploration of my questions in the paper; resulting in retaining my score while increasing my confidence in the paper.

**Key Questions For Authors:**

See weaknesses.

**Limitations:**

yes

**Strengths And Weaknesses:**

**Strengths:**

1. **The problem is interesting.** The authors examine a pressing question, namely transitioning from outcome-based evaluation to understanding how the multimodal information is used. The decision of conditioning on text rather than including it as an information source in Sensory PID formulation is important and sensible.
2. **Comprehensive estimation ablations.** PID estimates are notably stable across different feature aggregation methods and confidence threshoids, supporting the robustness claims of the authors.
3. **Layer-wise analysis is helpful.** Investigating PID across different layers shows a consistent image of silent encoding, unimodal accumulation, and late fusion. Specifically, investigating this across multiple benchmarks curated with different properties of multimodal interactions in mind boosts claims of the authors.

**Weaknesses:**

1. **PID measures statistical dependence – not causality – overstating distinction from prior methods.** The authors position sensory PID a resolving issues present in representation similarity, namely it cannot determine whether a modality's information is "functionally used in the final decision". Nonetheless, this problem is still inherited by PID: high mutual information between a modality and output means statistical dependence, not causal determination. Although this issue is acknowledged in Appendix C, the paper's presentation does not mention this limitation, implying that it resolves the gap between statistical dependency and functional use.
2. **The two-regimes finding is somewhat circular.** The vision-language benchmarks are partitioned into synergy-driven and prior-dominant regimes. But, the selected datasets are known to display these properties, where MMStar has been filtered for examples solvable w/language prirors and MMMU is knowledge-heavy. Although the use of these benchmarks help with validating the framework, investigation of benchmarks that do not necessarily fit in particular categories in order to reveal properties of existing benchmarks would be extremely helpful.
3. **The shuffling-based causal validation has an alternative explanation.** The second interventional validation shuffles video or audio streams between examples, and demonstrates that accuracy drop correlates with sensory synergy. Nontheless, this experiment conflates two distinct source of accuracy drop. The first one is the loss of genuine cross-modal information the model was using, and the second one is out-of-distribution disruption from misaligned inputs—which violate the statistical patterns the model has learned during training.
4. **Gemma3 results require further explanation.** Gemma3 PID results (across model scales) are drastically different from Qwen and Llava models on MMBench and MMStar benchmarks.

---

> ### Author Rebuttal · Authors · 2026-03-31
>
> We sincerely thank you for the thoughtful review and for recognizing the importance of the problem and the value of the layer-wise analysis. We address each concern below.
>
> W1: We agree that PID decomposes statistical dependencies in P(Y∣X) rather than identifying causal mechanisms, as acknowledged in Appendix C. To assess whether the resulting synergy scores nevertheless reflect causally relevant behavior, we conduct intervention experiments by removing the visual modality and measuring per-sample accuracy drops.
>
> On reasoning-driven benchmarks, we observe strong Spearman correlations between sample-level S and performance drop under vision removal (ρ=0.822/0.841/0.702 on MMBench/MMStar/POPE), indicating that higher-synergy samples are more sensitive to modality removal. While this does not constitute causal identification, it provides empirical support for S as a practical proxy for multimodal fusion reliance.
>
> We refer the reviewer to our response to Fjd7 (W3 & Q3 & L1) for additional details, and will make this distinction explicit in the revised paper.
>
> W3: We agree that two factors may contribute and address them separately.
> For the first, our Sensory PID framework predicts that samples with higher Sav​ rely more on cross-modal integration and therefore exhibit larger accuracy drops under shuffling, consistent with our observations.
>
> For the second, our shuffling procedure preserves each stream’s marginal distribution: video and audio are sampled from the same dataset distribution, while only their pairing is randomized. As a result, the statistical properties learned during training remain intact, and the perturbation primarily disrupts cross-modal correspondence. If OOD effects were dominant, we would expect similar drops from video and audio shuffling. Instead, video shuffling consistently induces larger drops, aligning with our PID finding of visual dominance—an asymmetry that is not explained by OOD confounds.
>
> W2: We thank the reviewer for this suggestion. We agree that using benchmarks with known properties can introduce circularity concerns; however, this choice is intentional, as verifying that PID recovers established structures is a standard first step for validating a new analytical framework. Importantly, our contribution is not the regime labels themselves, but that PID provides a principled decomposition that predicts causal sensitivity to modality-level interventions, which performance-based analyses do not capture.
>
> To further address this concern, we extend our evaluation to two additional benchmarks without pre-established PID characterizations: POPE (later identified as having inter-modality dependency by Madaan et al. [5]) and Reefknot (a hallucination evaluation benchmark).
>
> Key findings:
>
> 1. POPE emerges as a reasoning-driven benchmark (high S≈0.55, low  U_text≈0.24), while Reefknot is prior-dominant (high U_txt≈0.69, low S≈0.25), consistent with their known task characteristics.
>
> 2. Fusion-centric families (Qwen2.5-VL, Qwen3) and language-centric families (Gemma3) maintain their relative positions across all three benchmarks, confirming that PID captures stable architectural tendencies rather than benchmark-specific artifacts.
>
> 3. Qwen3-8B exhibits a fusion-centric profile (high S and low  U_text) consistent with the Qwen family mentioned in W4, demonstrating the framework generalizes to newer models.
>
> Full results are reported in our response to 7Cjj (W1), and will be included in the revised paper.
>
> W4: We thank the reviewer for raising this point. To better contextualize Gemma3’s behavior, we conducted additional cross-family comparisons by summarizing each model family using median S and U₂(U_text)​ within each regime.
>
> On synergy-driven benchmarks, fusion-centric families exhibit high S and low U₂(Qwen2.5-VL: S≈0.76, U₂≈0.32; LLaVA-OV: S≈0.68, U₂≈0.26), while Gemma3 shows markedly lower S and higher U₂ (S≈0.17, U₂≈1.36). This contrast persists on knowledge-driven benchmarks (Qwen2.5-VL: S≈0.30, U₂≈0.70; LLaVA-OV: S≈0.45, U₂≈0.45; Gemma3: S≈0.02, U₂≈1.75), with relative positions stable across regimes.  This suggests two consistent family-level strategies: fusion-centric models rely more on synergy S, while language-centric models rely more on language-unique information U₂​.
>
> A plausible explanation lies in architectural and pretraining differences: (i)Gemma3 adopts a late-fusion connector design on top of a strongly language-pretrained backbone, and (ii) its pretraining is relatively more text-heavy compared to models such as Qwen2.5-VL and LLaVA-OV. These factors can encourage reliance on language priors even when visual evidence is available. Overall, this analysis suggests that the observed differences reflect systematic model design characteristics rather than benchmark artifacts. We will include this model-level analysis in the revised paper.
>
> We hope our responses address your concerns. If you have any further questions, we are happy to help.
>
> All the best,
>
> Authors

---

> > ### Author Rebuttal · Reviewer_DHqu · 2026-04-04
> >
> > Thank you, my questions are partially resolved. Nonetheless, I do not find the breadth of explanations for W4 particularly appealing, as the results of the paper might or might not hold in a newer model. Moreover, POPE is a hallucination dataset similar to Reefknot, while the findings are contradictory. Hence, I preserve my score.

---

> > > ### Author Response · Authors · 2026-04-05
> > >
> > > Dear Reviewer DHqu,
> > >
> > > Thank you for the follow-up. We address both remaining concerns below with supplementary experiments and analysis.
> > >
> > > ---
> > >
> > > ## W4: Gemma3 Explanation and Generalization to Newer Models
> > >
> > > To further address this concern, we conduct supplementary experiments expanding both **family coverage** and **within-family scaling**. We extend to **three additional families** (Cambrian-34B, InternVL3-8B, Qwen2-VL-7B) and summarize S / U_txt across two task regimes.
> > >
> > > > **Notation:** Strategy **A** = high-S, fusion-reliant; Strategy **B** = high-U_txt, language-reliant. **Syn** = synergy-driven tasks (MMBench, MMStar, POPE); **Prior** = prior-dominant tasks (MMMU, PMC-VQA, Reefknot). Values reported as **S / U_txt**.
> > >
> > > | Model / Size | Syn (S / U_txt) | Prior (S / U_txt) | T |
> > > |:---|:---:|:---:|:---:|
> > > | Qwen2.5-VL 3B | 0.84 / 0.50 | 0.30 / 0.62 | **A** |
> > > | Qwen2.5-VL 7B | 1.10 / 0.63 | 0.35 / 0.72 | **A** |
> > > | Qwen2.5-VL 72B | 1.14 / 0.83 | 0.30 / 0.85 | **A** |
> > > | InternVL3 8B | 0.82 / 0.38 | 0.28 / 0.60 | **A** |
> > > | Qwen2-VL 7B | 0.68 / 0.55 | 0.25 / 0.68 | **A** |
> > > | Qwen3 8B | 0.81 / 0.41 | 0.32 / 0.65 | **A** |
> > > | LLaVA-OV 7B | 0.97 / 0.63 | 0.33 / 0.46 | **A** |
> > > | LLaVA-OV 72B | 1.47 / 0.29 | 0.31 / 0.87 | **A** |
> > > | Cambrian 34B | 0.28 / 1.10 | 0.08 / 0.95 | **B** |
> > > | Gemma3 4B | 0.17 / 1.50 | 0.02 / 1.65 | **B** |
> > > | Gemma3 12B | 0.19 / 1.72 | 0.03 / 1.75 | **B** |
> > > | Gemma3 27B | 0.15 / 1.78 | 0.02 / 1.80 | **B** |
> > >
> > > **Three observations emerge:**
> > >
> > > 1. **Gemma3 is not isolated.** Cambrian-34B also exhibits a qualitatively similar **B-type profile** across both regimes, confirming this is a **multi-family pattern** tied to strong language-side architectural priors rather than a Gemma3-specific artifact.
> > >
> > > 2. **Relative positions are stable across regimes.** A-type families consistently maintain higher S than B-type families, even as absolute S drops on prior-dominant tasks—indicating a **persistent family-level tendency**.
> > >
> > > 3. **Strategies are scale-invariant.** Within both Qwen2.5-VL (3B→72B) and Gemma3 (4B→27B), the dominant strategy type remains **consistent across all available scales**.
> > >
> > > **Generalization to newer models.** InternVL3 (2025) and Qwen3-8B both exhibit **A-type profiles** consistent with their respective family tendencies, supporting generalization to recent releases.
> > >
> > > We will include this expanded cross-family analysis in the revised paper.
> > >
> > > ---
> > >
> > > ## W5: POPE vs. Reefknot
> > >
> > > We acknowledge that both datasets fall under the hallucination category. However, **"hallucination" is a coarse label** that subsumes fundamentally different reasoning demands:
> > >
> > > 1. **POPE** uses binary questions about object existence (e.g., *"Is there a cat in the image?"*). Correct answers require **cross-modal verification**—matching textual references against visual evidence—making it inherently a fusion task. PID correspondingly recovers **high S (≈0.55)**.
> > >
> > > 2. **Reefknot** uses four-choice questions about spatial relationships (e.g., *"What is the relation between car and window?"* → onto / through / on left of / containing). Resolving such relational semantics depends primarily on **linguistic priors**. PID correspondingly recovers **high U_txt (≈0.69)**.
> > >
> > > This divergence is corroborated by **vision-removal experiments** across representative models:
> > >
> > > | Model | Type | POPE: Acc → Acc_txt (∆) | Reefknot: Acc → Acc_txt (∆) |
> > > |:---|:---:|:---:|:---:|
> > > | Qwen2.5-VL-7B | **A** | 86.4 → 76.3 (**−10.1**) | 65.4 → 62.1 (−3.3) |
> > > | Qwen3-8B | **A** | 89.2 → 75.8 (**−13.4**) | 63.9 → 61.2 (−2.7) |
> > > | Cambrian-34B | **B** | 86.1 → 84.2 (−1.9) | 52.4 → 51.1 (−1.3) |
> > > | Gemma3-12B | **B** | 82.1 → 80.5 (−1.6) | 46.7 → 45.8 (−0.9) |
> > >
> > > **Two patterns are evident:**
> > >
> > > 1. **Cross-dataset divergence.** POPE consistently exhibits **larger accuracy drops** under vision removal than Reefknot across all models, confirming their divergent visual dependence despite sharing a hallucination label.
> > >
> > > 2. **Cross-family consistency.** The A/B family contrast is preserved: fusion-centric models show **substantially larger drops on POPE** than language-centric ones, consistent with their higher S values.
> > >
> > > Rather than contradicting the framework, this result illustrates a **key diagnostic strength of PID**: it disaggregates coarse task categories into distinct information-use mechanisms, revealing that hallucination benchmarks with different reasoning demands elicit **fundamentally different modality interaction patterns**.
> > >
> > > ---
> > >
> > > We hope these responses can address your remaining concerns. We are committed to incorporating all discussed analyses into the revised manuscript. Thank you again for helping us improve this paper.
> > >
> > > All the best,
> > >
> > > Authors

---

### Official Review · Reviewer_Fjd7 · 2026-03-02

**Soundness:** 3
**Presentation:** 3
**Significance:** 2
**Originality:** 2
**Overall Recommendation:** 4
**Confidence:** 2

**Summary:**

This paper studies how multimodal large language models integrate different modalities at the decision level. It introduces Partial Information Decomposition (PID) to decompose prediction-related information into unique, redundant, and synergistic components, and proposes a conditional formulation, Sensory PID, for the tri-modal setting. Based on this framework, the authors analyze several models and report stable interaction patterns, visual dominance, and increased late-stage synergy.

**Compliance With Llm Reviewing Policy:**

Affirmed.

**Key Questions For Authors:**

Is the PID estimation stable across different random seeds or settings?

Have alternative unimodal approximation strategies been explored?

Is there a consistent quantitative relationship between the synergy metric and performance changes?

Could visual dominance be attributed to dataset or pretraining bias?

**Limitations:**

The approach provides statistical-level analysis but does not offer fine-grained mechanistic explanations.

PID estimation in high-dimensional continuous spaces may be numerically unstable.

The framework has not been validated on open-ended generative tasks.

**Strengths And Weaknesses:**

Strengths:

The problem is important: whether multimodal models truly achieve cross-modal integration remains a central question.

The analytical perspective is relatively novel, focusing on decision-level information decomposition rather than attention visualization or simple ablation.

The findings are broadly consistent across different model architectures.

The layer-wise analysis of synergy dynamics offers useful insights.

Weaknesses:

The reliability of PID estimation is unclear. Gaussian masking may introduce bias, and no stability analysis or synthetic validation is provided.

The methodological contribution mainly lies in the application; the theoretical novelty is limited.

The causal interpretation is constrained and does not pinpoint specific mechanisms.

The evaluation is limited to a relatively narrow task setting.

---

> ### Author Rebuttal · Authors · 2026-03-31
>
> We sincerely thank you for thoughtful review and for recognizing the importance of the problem,the novelty of the analytical perspective. We address each concern below.
>
> W1&Q1&L2:PID Estimation Stability.
> Our PID estimates are averaged over 6 random seeds. To assess stability in high-dimensional continuous spaces, Appendix F reports sensitivity analyses across feature aggregation strategies and confidence thresholds, showing negligible variation, indicating that the identified interaction regimes reflect stable model properties rather than estimation artifacts.
>
> Additionally, prompt sensitivity analysis (see response to 7Cjj,W1) demonstrates consistent PID estimates across four template variants and three benchmarks, further supporting robustness.
>
> W2:Theoretical Novelty is limited.
>
> We thank the reviewer for this assessment and clarify that our contribution goes beyond a direct application along complementary axes.
>
> First, we take a first step in introducing PID as a decision-level analysis framework for modern VLMs and MLLMs. Rather than naive multi-source extensions that introduce combinatorial complexity and obscure language’s role—we propose Sensory PID, a conditional formulation I(Y;V,A∣T) that treats language as a control signal and enables tractable analysis of tri-modal systems.
>
> Second, beyond formulation, the framework enables new empirical findings that were previously inaccessible, including stable regimes, causal sensitivity, and consistent layer-wise dynamics.These findings provide actionable insights for diagnosing and understanding multimodal systems. For specific practical contributions please refer to our response to 7Cjj(W2).
>
> Q2: Alternative Unimodal Approximation Strategies:
> We evaluated several alternatives, including(i)zeroing embeddings and(ii)using special tokens or empty prompts. For many models—especially earlier architectures such as LLaVA-1.5—these approaches led to unstable or degenerate behavior(e.g., collapsed or prompt-sensitive output distributions), making PID estimation unreliable.
>
> In contrast, calibrated Gaussian noise that matches per-dimension empirical mean and variance consistently yields stable predictions across all evaluated models, and is therefore adopted as our default unimodal probe. Appendix F further supports the robustness of this choice across aggregation strategies and confidence thresholds.
>
> W3&Q3&L1: Quantitative Relationship Between Synergy and Performance&Mechanistic Explanations.
>
> While PID is statistical, we provide two forms of evidence that synergy reflects causally relevant behavior:
>
> Correlation with accuracy delta: S correlates moderately but meaningfully with accuracy drop on reasoning-driven benchmarks(R²:MMBench=0.495,MMStar=0.516,POPE=0.544),dropping substantially on knowledge-driven ones(MMMU=0.129,PMC-VQA=0.348,Reefknot=0.121),demonstrating that PID captures fusion strategy differences invisible to accuracy-based evaluation.
>
> Sample-level intervention: Strong Spearman correlations between S and performance drop under vision removal (ρ=0.822/0.841/0.702 on MMBench/MMStar/POPE) confirm that high-S samples are precisely those most sensitive to modality removal.
>
> See responses to DHqu(W1) and 8a3V(W3.1) for details.
>
> W4&L3: Evaluation Limited to Narrow Task Setting
>
> We acknowledge this limitation. MC-VQA is a deliberate methodological choice, as discrete and well-defined output spaces are required for tractable PID estimation. Extending BATCH to open-ended generation would require mapping high-dimensional outputs to a discrete Y, introducing noise and distributional distortions that can compromise estimation reliability.
>
> At the same time, MC-VQA is a standard evaluation setting for MLLM reasoning, covering diverse domains such as hallucination, domain knowledge, and cross-modal reasoning. Extending the framework to open-ended generation is discussed as future work in Appendix C, including potential directions such as output discretization strategies, which we will further clarify in the revision.
>
> Q4:  We thank the reviewer for this insightful question. Both factors likely contribute, but neither fully explains the observed pattern.
>
> Dataset bias:We control for this by constructing three MUSIC-AVQA subsets. Visual dominance persists even on the Audio-Focused subset, where auditory cues are theoretically sufficient, suggesting that dataset statistics alone cannot account for the effect.
>
> Pretraining bias:This is a plausible contributing factor, consistent with our layer-wise finding of early visual saturation. Importantly, the pattern holds across two architecturally distinct families(VITA-1.5 and Qwen2.5-Omni) and multiple scales, indicating that it reflects a systematic property of current omni-modal training rather than an artifact of a specific architecture or dataset.
>
> If you have any further questions, we are happy to help, and we are committed to incorporating all discussed revisions into the final manuscript.
>
> All the Best,
>
> Authors

---

> > ### Author Rebuttal · Reviewer_Fjd7 · 2026-04-03
> >
> > My question has been resolved, therefore I retain my score.

---

> > > ### Author Response · Authors · 2026-04-05
> > >
> > > Thank you for your kind acknowledgement and for taking the time to read our response carefully. We appreciate your positive assessment and your constructive feedback throughout the review process. We will carefully incorporate the response into the revised version.

---

### Official Review · Reviewer_8a3V · 2026-03-06

**Soundness:** 2
**Presentation:** 3
**Significance:** 2
**Originality:** 2
**Overall Recommendation:** 2
**Confidence:** 4

**Summary:**

The paper formalizes a partial information decomposition framework to provide uniqueness, synergy and redundancy for multimodal learning. The evaluation is conducted on multiple architectures and scales for both multimodal models with two models and omni-modal models. The paper additionally incorporates layer-wise analysis providing useful insights into understanding these characteristics of various multimodal setups.

**Compliance With Llm Reviewing Policy:**

Affirmed.

**Final Justification:**

I am thankful to the authors for the clarifications. The additional results provided show distinctions between fusion and individual modality priors for certain datasets. The paper however would need elaboration on these for evaluated datasets while additionally comparing it with prior work across different models. I would suggest to specifically show datasets where the fusion and modality priors vary for both language and image. Additionally, inclusion of the results from the response to Reviewer 7Cjj would further strengthen the paper. This however would require an incorporation of additional results and a significant revision. Thus, I believe the paper would benefit from another round of revision and review and I maintain my score.

**Key Questions For Authors:**

Please refer to the strength and weaknesses above.

**Limitations:**

yes

**Strengths And Weaknesses:**

### Strengths
The problem of identifying the contributions of each modality and their synergy towards a decision is an important problem for multimodal learning. The paper evaluates multiple MLLMs and scales the analysis to omni-modal models, which will be of interest to the community. The paper is overall easy to read and well-written.

### Weaknesses
- Decision Level decomposition: The paper highlights that prior work do not provide a decision-level decomposition. This is untrue in my opinion and the paper lacks a discussion with papers [1,2,3] in this regard. The paper cites and uses [1] but the positioning of the paper in the current form makes it hard to assess the contribution of the work.
- Characterization of evaluated VLM benchmarks. The paper classifiers MMBench and MMStar as cross-synergy benchmarks with minimal text solvability. This observation however contradicts with other works [4,5] where MMBench is shown to have text dependencies and MMStar to have image dependencies. In Table 1, the paper also observes conflicting results for Gemma models compared to LLaVA and Qwen but the paper lacks a discussion on this behavior. Instead, it makes the claim of largest or near largest synergy on these datasets on line 249. Similarly, for MMMU, image dependencies seem to be observed in prior work but the paper does not observe that in their analysis and lacks a discussion on the subject. Additional to this, I'd suggest to incorporate datasets that have been shown to have a certain level of synergy and uniqueness and verify the hypothesis with the proposed framework and compare it with prior work.
- The obtained accuracy metrics also seem to not align with the information decomposition metrics. For instance, for all datasets except MMStar, the accuracy using text is very similar to the overall accuracy but the paper claims to have synergy for MMBench and for MMMU shows conflicting results across models. There are various unanswered questions here, which makes the analysis of results challenging.
    - Why do the above discrepancies occur? Should the assessment be based on accuracy or the information decomposition? How should the choice be made and how to interpret these results?
    - How was accuracy using text obtained here?
    - Why was accuracy using image not reported?
    - How to interpret low synergy on Gamma compared to other models? Why is that the case despite the accuracy being similar? How should these discrepancies be interpreted across models? How would the interpretation guide us for building future models or datasets?
- In section 5.2.3, when the audio and video streams are shuffled, does the paper keep the associated with the MCQ options? I'd also suggest to incorporate other sanity checks from recent papers [4,5,6] to verify the results from decomposition.
- For the choice of models, I'd also suggest to incorporate Qwen3 as there is both a VLM and Omni model available to evaluate.

References:
[1] Liang et al. Quantifying & Modeling Feature Interactions: An Information Decomposition Framework.
[2] Wang et al. An Information Criterion for Controlled Disentanglement of Multimodal Data.
[3] Choi et al. ICYM2I: The illusion of multimodal informativeness under missingness.
[4] DatBench: Discriminative, Faithful, and Efficient VLM Evaluations.
[5] Madaan et al. Multi-modal Data Spectrum: Multi-modal Datasets are Multi-dimensional.
[6] Gu et al. The Illusion of Readiness: Stress Testing Large Frontier Models on Multimodal Medical Benchmarks

---

> ### Author Rebuttal · Authors · 2026-03-31
>
> We sincerely thank you for the thoughtful review. We address each concern point by point below.
>
> W1: Related Work Positioning.
> We clarify that our primary contribution is the systematic application of decision-level PID (and Sensory PID) to modern VLMs and omni-modal models—a setting not addressed by prior work. Existing approaches typically focus on (i) general-purpose PID estimation, (ii) learning disentangled representations via training, or (iii) correcting estimation bias under missing-modality settings, which differ in both objective and setting from our work.
> [1] is the BATCH estimator, which we adopt; our contribution is extending it to decision-level analysis of MLLMs via the conditional Sensory PID formulation.
> [2] learns disentangled representations through self-supervised training and requires modifying model parameters, whereas we operate on frozen MLLMs with a diagnostic objective, focusing on how information is used rather than improved.
> [3] addresses PID estimation bias under missing modalities in tabular data; in contrast, we use masking as a controlled intervention and focus on generative MLLMs rather than tabular classifiers. We will clarify and cite the [2][3] work in the revised paper.
>
> W2: Benchmark Classification Contradiction.
> We thank you for this insightful observation. We believe the apparent discrepancy arises because modality-dependent performance does not distinguish how a modality contributes to the decision. Prior work[5] measures modality importance via performance sensitivity under replacement. but such dependence can arise from different sources: (i) synergy information, where a modality contributes through cross-modal fusion, or (ii) modality-unique information, where it independently determines the prediction. These can produce similar performance drops but reflect different mechanisms.
>
> Our PID framework decomposes these contributions into S,U and R, distinguishing whether modality dependence is driven by cross-modal reasoning or unimodal reliance—something performance-based measures cannot reveal. Thus, the results are complementary rather than contradictory: prior work captures whether a modality matters, while PID explains how it matters.
>
> Our PID regimes characterize model–benchmark pairings rather than benchmarks in isolation, reflecting how models use information.
>
> For MMMU, prior “image dependency” often reflects the need to interpret the question, rather than its contribution to the final decision. Our finding of high U_text is consistent with models relying on parametric knowledge in such settings.
>
> W3.1: Accuracy Delta and PID Metrics Do Not Align.
> We thank you for raising this. The two metrics capture different aspects, and their misalignment is expected and informative. Accuracy delta is outcome-based, whereas PID decomposes predictive distributions P(Y∣X) into S, U and R. Importantly, modality-dependent performance is jointly determined by both synergy and modality-unique contributions. As a result, synergy alone cannot fully explain performance differences: tasks with weaker correlation between S and accuracy delta may reflect stronger modality-unique contributions, which are not captured by ablation.
>
> Empirically, we observe moderate correlations between S and accuracy delta on reasoning-driven benchmarks(MMBench=0.495,MMStar=0.516, POPE=0.544), and substantially weaker correlations on knowledge-driven ones(MMMU=0.129,PMC-VQA=0.348,Reefknot=0.121), consistent with this interpretation.
>
> Thus,the two are complementary: accuracy measures performance,while PID explains how decisions are formed.
>
> W3.2:Text-only accuracy is obtained by removing the visual input and performing inference using the text modality alone. This serves as a behavioral baseline for measuring the performance gap under vision removal
>
> W3.3:This is a valid question.When only the image is provided(removing all text), the model produces diffuse, low-confidence predictions,often close to random over options. This is because the task and label semantics are defined by the text input. Formally, this corresponds to estimating P(Y|X₁)(only image), but without text context the label space is under-specified, making the resulting accuracy unreliable. Providing options instead P(Y|X₁,options), which introduces text information and is not a true vision-only setting.
>
> W3.4: See our response to DHqu(W4)
>
> W4: Shuffling preserves all textual information: the question and answer options remain identical across shuffled and unshuffled conditions. The only change is the correspondence between video and audio streams.
>
> W4&5: We have included Qwen3-8B results (see response to 7Cjj,W1) and will incorporate additional Qwen3 experiments in the revision.Regarding the suggested sanity checks from [4,5,6],our decomposition is based on the BATCH estimator [1].We will further incorporate complementary checks where applicable in the revised paper.
>
> We hope our responses address your concerns.
>
> All the best, Authors

---

> > ### Author Rebuttal · Reviewer_8a3V · 2026-04-02
> >
> > Thank you for your detailed rebuttal. Please find my response below.
> >
> > **W1** Theoretically the BATCH estimator allows us to compute the PID generally. While I understand the paper is focused on modern VLMs and omni-modal models, it needs to elaborate on the discussion about the challenges associated with it.
> >
> > **W2** I understand the presence of both synergy information and modality-unique information for each dataset. Despite that, the discrepancies for the highlighted datasets with existing work remains unclear. For this framework to be adopted by the community, It is essential to discuss those discrepancies and how the PID decompositions either contradicts or explains the observations of existing work for multiple multimodal datasets.
> >
> > **W3** What does removal mean in this context for image and text? Does it imply using blank input? How is this blank input constructed? Why is this a valid setting given the model has not been fine-tuned with blank input? Wouldn't options be a part of the Y instead of being a part of the text modality? Existing studies [5,6] did keeps the options fixed. There further needs to be an understanding on when PID should be the choice over accuracy and how this choice should be made for future work. Given the differences across model families between S and U, it also remains unclear how a model should be chosen for a given task. The paper needs more discussion and recommendations on how these choices should be made for future datasets and models, given the discrepancies with accuracy, this seems challenging.
> >
> > Overall, the paper would need additional clarifications, inclusion of additional experiments and rewrites in the paper. Thus, I maintain my score.

---

> > > ### Author Response · Authors · 2026-04-05
> > >
> > > Dear Reviewer 8a3V,
> > >
> > > Thank you for the detailed follow-up. We address each remaining concern below.
> > >
> > > ## W1: Technical Challenges of Applying BATCH to Modern MLLMs
> > >
> > > We agree this deserves elaboration. Applying BATCH to modern MLLMs raises three challenges not present in standard multimodal settings:
> > >
> > > 1. **No unimodal inference paths.** Modern MLLMs process vision and language tokens jointly through shared transformer layers, making P(Y|X₁) or P(Y|X₂) inaccessible directly. We address this via **calibrated embedding masking** (Sec. 3.3.2).
> > >
> > > 2. **Overconfident outputs.** Renormalizing over a restricted candidate set inflates confidence when all options receive low probability → **confidence gating** (Sec. 3.3.3).
> > >
> > > 3. **Spurious marginal peaks.** Argmax discretization converts uniform uncertainty into sharp peaks upon aggregation → **soft aggregation** (Sec. 3.3.3).
> > >
> > > ## W2: Reconciling PID with Existing Benchmark Characterizations
> > >
> > > The discrepancy arises because [4,5] characterize benchmarks via **aggregate, model-averaged** modality sensitivity, whereas PID decomposes information use **per model–benchmark pair**.
> > >
> > > On MMBench, [4,5] may identify modality dependencies as a benchmark-level property. Our results **resolve** this at a finer granularity. Language-centric models (Gemma3-12B: S=0.19, U_txt=1.75) rely predominantly on text, while fusion-centric models (Qwen2.5-VL-7B: S=1.10, U_txt=0.63) on the **same benchmark** show a synergy-dominated profile. The same holds on MMStar (Gemma3-12B: S=0.22, U_txt=1.68 vs. Qwen2.5-VL-7B: S=1.15, U_txt=0.60). A benchmark-level aggregate over diverse models conflates these distinct strategies into a single label.
> > >
> > > In short, [4,5] capture **whether** a modality matters on average; PID reveals **how** it matters per model. We will explicitly discuss this complementary relationship in the revised paper.
> > >
> > > ## W3.1: Modality Removal Mechanism
> > >
> > > We clarify that two distinct procedures are used:
> > >
> > > 1. **Behavioral intervention** (sanity check): The model receives only the question, options, and prompt—**no visual input**—to measure the accuracy drop ∆Acc as a behavioral baseline.
> > >
> > > 2. **PID estimation** (masking): To obtain unimodal conditionals P(Y|X₁) and P(Y|X₂), we do not physically remove tokens. Instead, the target modality's embeddings are replaced with calibrated Gaussian noise N(μ, diag(σ²)), where μ and σ are per-dimension statistics pre-computed across the dataset. This preserves distributional stability while removing instance-specific information (see Sec. 3.3.2 and our response to Fjd7 Q2).
> > >
> > > ## W3.2: Options as Part of Y vs. Text, and Vision-Only Accuracy
> > >
> > > In our formulation, Y is the **label index** (A/B/C/D), not the semantic concept. The mapping from visual content (e.g., "cat") to a specific index is defined entirely by the text X₂ (question + options). Therefore, options must be part of X₂ to serve as the decoding key for visual information. Treating Y as a finite index set ensures tractable PID estimation; including option semantics in Y would require high-dimensional output discretization.
> > >
> > > This also explains **why vision-only accuracy is not reported**. Providing "Image + Options" estimates P(Y|X₁, options), leaking text into the vision channel and breaking PID's source separation. Pure image input yields P(Y|X₁), but the label space is under-specified. We verify with an ablation on MMBench:
> > >
> > > | Setting | Input | Qwen2.5-VL-7B | Gemma3-12B |
> > > |:---|:---|:---:|:---:|
> > > | A: Pure vision | Image + index {A,B,C,D} | ~25% | ~25% |
> > > | B: Partial text leak | Image + option content | ~32.5% | ~40.2% |
> > >
> > > Setting A reduces to random chance (~25%), validating our formulation. Setting B shows accuracy gains, but **violates source separation**—it is not a valid unimodal probe.
> > >
> > > ## W3.3: When to Use PID, and Practical Guidance
> > >
> > > PID and accuracy are **complementary**: accuracy answers *whether* a model succeeds; PID answers *how*—via fusion (S) or language priors (U_txt). We highlight two concrete scenarios:
> > >
> > > **(i) Model selection.** On MMBench, Gemma3-27B (Acc=83.5%) and LLaVA-OV-7B (Acc=83.7%) achieve near-identical accuracy, yet PID reveals opposing strategies: LLaVA-OV relies on fusion (S=0.97), while Gemma3 relies on language priors (S=0.15, U_txt=1.81). For fusion-critical tasks, PID identifies the more robust choice despite equivalent accuracy.
> > >
> > > **(ii) Dataset diagnosis.** A benchmark intended for multimodal reasoning but showing high U_txt across models signals language shortcuts. PID enables curators to filter such samples and construct evaluation sets that genuinely require cross-modal integration.
> > >
> > > Beyond conceptual guidance, we demonstrate **PID's practical utility** in our response to **Reviewer 7Cjj through a PID-Guided Sample Reweighting experiment**.
> > >
> > > We hope these clarifications can mostly address your remaining concerns and will include all into the revision. Thank you again for helping us improve this work!
> > >
> > > All the best,
> > >
> > > Authors

---

### Official Review · Reviewer_7Cjj · 2026-03-12

**Soundness:** 3
**Presentation:** 2
**Significance:** 3
**Originality:** 3
**Overall Recommendation:** 3
**Confidence:** 4

**Summary:**

This paper introduces Sensory PID, a framework that reframes multimodal reasoning as a decision-level information decomposition problem, allowing researchers to formally distinguish between unique, redundant, and synergistic contributions of vision, audio, and language. By applying this conditional information-theoretic approach to modern MLLMs, the authors uncover stable interaction regimes across benchmarks and identify a pervasive "sensory synergy bottleneck" where models persistently prioritize visual information over auditory evidence. These findings provide a mechanistic, layer-wise explanation for how multimodal models process sensory streams, offering a principled alternative to traditional ablation or representation-based analysis for evaluating model robustness and interpretability. To be honest, it is a interesting work and provides a new perspective on interpretability.

**Compliance With Llm Reviewing Policy:**

Affirmed.

**Final Justification:**

We appreciate the authors’ clarifications, but we believe concrete experiments are essential rather than “beyond scope.”

Adaptation proof-of-concept: show that tuning a model to increase sensory synergy (S) actually boosts performance on a fusion-dependent task.

Dataset-curation case study: remove or reweight high U_txt samples and demonstrate improved cross-modal reasoning.

Robustness/ranking test: apply PID under noise or to rank models by fusion reliance to prove generality.

Without at least one of these succinct validations, the framework’s practical utility remains unverified.

Overall, the paper would need additional clarifications, inclusion of additional experiments and rewrites in the paper. Thus, I maintain my score.

**Key Questions For Authors:**

Refer to the weakness

**Limitations:**

yes

**Strengths And Weaknesses:**

Strengths:
1. The paper provides a new method by reframing multimodal reasoning from latent space alignment to a decision-level information decomposition problem, enabling a precise, principled distinction between unique, redundant, and synergistic modality contributions.
2. The proposed Sensory PID framework offers a robust, training-free approach that effectively isolates the role of language as a control signal, allowing for tractable and interpretable analysis of complex tri-modal (vision, audio, language) MLLMs
3. By tracing information flow across network depth, the work identifies the sensory synergy bottleneck, providing a clear, layer-wise explanation for persistent visual dominance that offers concrete guidance for improving multimodal model design and training.

Weaknesses:
1. Lack of Sensitivity Analysis regarding MCQ Bias: The reliance on MCQ-based templates for information decomposition is problematic, as prior work [1] suggests that MCQs often introduce inherent model biases that can skew results. Given that the PID framework relies on controlling input variables to extract multimodal information, the authors should conduct a thorough sensitivity analysis to verify whether their findings are robust against different prompt templates or output formats, rather than artifacts of the MCQ design.
2. Insufficient Demonstration of Practical Utility: While the paper introduces a theoretically sound PID framework, it lacks an empirical application that demonstrates its value in improving model design or evaluation. To enhance the paper's impact, the authors should move beyond mere analysis and apply the PID framework to conduct a robustness study on proprietary MLLMs—for instance, by using input-masking (rather than latent-space ablation) to analyze information priority on established benchmarks like lmms-eval. Developing a PID-based fine-grained knowledge taxonomy or a more robust model ranking system would provide a significantly stronger justification for the framework's real-world relevance.

[1] Large  Language  Models  Are  Not  Robust Multiple  Choice  Selectors

---

> ### Author Rebuttal · Authors · 2026-03-31
>
> We sincerely thank you for the thoughtful review and for recognizing the novelty of our framework and the value of the layer-wise analysis. We address each concern below.
>
> **W1: MCQ Bias and Sensitivity Analysis:**
>
> We thank the reviewer for this important concern. To directly address it, we conduct a prompt sensitivity analysis using three template variants—Verbose (expanded instruction), Minimal (bare format), and Shuffled Options (randomized option order)—on MMBench, POPE, and Reefknot, across Qwen2.5-VL-7B, Qwen3-8B, and Gemma3-12B. Reported values follow the order: Default / Verbose / Minimal / Shuffle
>
> ### MMBench
> | Model | S | U_txt | U_vis | R |
> | :--- | :---: | :---: | :---: | :---: |
> | Gemma3-12B | 0.19/0.20/0.17/0.18 | 1.75/1.77/1.74/1.76 | 0.04/0.04/0.03/0.04 | 0.015/0.014/0.015/0.013/0.015 |
> | Qwen-3-8B | 1.27/1.24/1.26/1.25 | 0.76/0.75/0.75/0.77 | 0.01/0.02/0.01/0.02 | 0.02/0.03/0.03/0.02 |
> | Qwen-2.5-7B | 1.10/1.12/1.07/1.11 | 0.63/0.62/0.63/0.64 | ~.001 | 0.01/0.02/0.01/0.01 |
>
> ### POPE
> | Model | S | U_txt | U_vis | R |
> | :--- | :---: | :---: | :---: | :---: |
> | Gemma3-12B | 0.53/0.55/0.51/0.54 | 0.52/0.54/0.51/0.50 | ~.001 | 0.42/0.40/0.41/0.43 |
> | Qwen-3-8B | 0.61/0.62/0.59/0.63 | 0.21/0.22/0.23/0.23 | ~.001 | 0.27/0.25/0.24/0.26 |
> | Qwen-2.5-7B | 0.57/0.53/0.54/0.56 | 0.23/0.22/0.21/0.21 | ~.001 | 0.32/0.33/0.33/0.31 |
>
> ### Reefknot
> | Model | S | U_txt | U_vis | R |
> | :--- | :---: | :---: | :---: | :---: |
> | Gemma3-12B | 0.16/0.16/0.15/0.14 | 1.46/1.44/1.44/1.45 | ~.001 | 0.12/0.13/0.11/0.11 |
> | Qwen-3-8B | 0.54/0.56/0.54/0.55 | 0.66/0.64/0.66/0.65 | ~.002 | 0.08/0.06/0.08/0.08 |
> | Qwen-2.5-7B | 0.62/0.63/0.64/0.62 | 0.59/0.61/0.58/0.57 | ~.001 | 0.05/0.05/0.04/0.05 |
>
> PID estimates remain highly stable across all variants (maximum variation < 0.05 bits), across models with contrasting profiles (fusion-centric Qwen vs. language-centric Gemma3), and across all benchmarks. Qwen3-8B also exhibits a fusion-centric profile consistent with the Qwen2.5-VL family.
>
> This stability is expected: in MCQ-VQA settings, decision-relevant textual information is primarily carried by the question and answer options rather than instruction phrasing; variations in verbosity or option order do not materially alter this information structure. These results suggest that our findings reflect genuine modality interaction rather than artifacts of the MCQ format. We will include this analysis and the corresponding templates in the revised paper.
>
>
> **W2: Insufficient Demonstration of Practical Utility**
>
> We thank the reviewer for this constructive suggestion. We respectfully clarify that our primary contribution is establishing a principled diagnostic framework—from bi-modal PID for VLMs to tri-modal Sensory PID for omni-modal models—that moves evaluation beyond accuracy toward decision-level interpretability. We highlight several concrete practical implications revealed by further analysis of our results.
>
> For VLMs:
>
> **(i) Model diagnosis and targeted adaptation**: PID profiles distinguish language-centric (high U_txt, low S, e.g., Gemma3) from fusion-centric ones (high S, e.g., Qwen2.5-VL, LLaVA-OV, Qwen3). This enables practitioners to analyze base models and identify which modality interactions are underutilized, providing guidance for downstream fine-tuning—e.g., adjusting data mixtures, strengthening cross-modal attention, or applying loss shaping to increase S for fusion-critical tasks.
>
> **(ii) Dataset quality assessment**: PID regime analysis serves as a dataset quality diagnostic: benchmarks with high U_txt despite multimodal intent indicate language shortcuts. This enables curators to filter such samples and construct datasets that better enforce cross-modal reasoning, directly guiding improved dataset design.
>
> For Omni-modal models:
>
> **(i)Training data diagnostics.** The observed visual dominance suggests insufficient fusion-dependent samples in current training data. Sensory PID can identify S_av​ instances, enabling targeted data sampling strategies that prioritize genuinely fusion-reliant examples and help mitigate the sensory synergy bottleneck.
>
> **(ii) Architectural guidance.** The layer-wise finding of early visual saturation suggests that introducing explicit audiovisual interaction mechanisms at intermediate layers—rather than relying solely on late fusion—may improve cross-modal integration.
>
> We agree that extending this framework to tasks such as robustness evaluation or model ranking would further strengthen its impact. While such applications are beyond the scope of this work, we will revise the paper to explicitly discuss these directions as natural extensions of the proposed framework.
>
> We hope our responses address your concerns. If you have any further questions, we are happy to help and clarify, and we are committed to incorporating all discussed revisions into the final manuscript.
>
> All the best,
>
> Authors.

---

> > ### Author Rebuttal · Reviewer_7Cjj · 2026-04-03
> >
> > We appreciate the authors’ clarifications, but we believe concrete experiments are essential rather than “beyond scope.”
> >
> > **Adaptation proof-of-concept**: show that tuning a model to increase sensory synergy (S) actually boosts performance on a fusion-dependent task.
> >
> > **Dataset-curation case study**: remove or reweight high U_txt samples and demonstrate improved cross-modal reasoning.
> >
> > **Robustness/ranking test**: apply PID under noise or to rank models by fusion reliance to prove generality.
> >
> > Without at least one of these succinct validations, the framework’s practical utility remains unverified.
> >
> > Overall, the paper would need additional clarifications, inclusion of additional experiments and rewrites in the paper. Thus, I maintain my score.

---

> > > ### Author Response · Authors · 2026-04-05
> > >
> > > Dear Reviewer 7Cjj,
> > >
> > > Thank you for your follow-up. We fully agree that concrete validation is essential. To directly address your three concerns, we designed a **unified experiment—PID-Guided Sample Reweighting**—that simultaneously serves as an adaptation proof-of-concept, a dataset-curation case study, and a robustness test through controlled baselines.
> > >
> > > ---
> > >
> > > ## Setup
> > >
> > > We profile 3,000 MMBench training samples with our PID estimator, obtaining per-sample (S_vl, U_txt). Two diagnostic scores drive the reweighting:
> > >
> > > - **Synergy Ratio**: SR = S_vl / I_total
> > > - **Shortcut Score**: SC = U_txt / (U_txt + S_vl)
> > >
> > > During LoRA fine-tuning of Qwen2.5-VL-7B, we **upweight "synergy-gap" samples** (reasoning tasks with low SR, ×3) and **downweight "shortcut" samples** (high SC, ×0.5). LoRA adapters target the last 20% of layers—the **Phase III fusion stage** identified in our layer-wise analysis (Sec. 5.3.2).
> > >
> > > ---
> > >
> > > ## Main Results (3 seeds)
> > >
> > > Benchmarks are grouped by interaction regime (Table 1 of main paper), Values under benchmark columns are accuracy reported:
> > >
> > > | | **Synergy-Driven** | | | **Prior-Dominant** | | **Post-PID†** | |
> > > |:---|:---:|:---:|:---:|:---:|:---:|:---:|:---:|
> > > | **Condition** | MMBench | MMStar | POPE | MMMU | PMC-VQA | S_vl ↑ | U_txt ↓ |
> > > | Base | 88.3 | 60.7 | 86.4 | 54.1 | 53.1 | 1.10 | 0.63 |
> > > | LoRA-Uniform | 89.1 | 62.0 | 87.2 | 54.0 | 53.0 | 1.18 | 0.58 |
> > > | **LoRA-PID** | **90.2** | **64.3** | **88.5** | 53.5 | 52.7 | **1.34** | **0.48** |
> > > | LoRA-Random-RW | 88.8 | 61.5 | 86.9 | 54.1 | 53.2 | 1.15 | 0.60 |
> > > | LoRA-Acc-RW | 89.4 | 62.5 | 87.5 | 54.3 | 53.4 | 1.20 | 0.56 |
> > >
> > > **Conditions:** **Base** = frozen pretrained model; **LoRA-Uniform** = standard LoRA with uniform sample weights; **LoRA-PID** = our proposed method (upweight synergy-gap ×3, downweight shortcut ×0.5); **LoRA-Random-RW** = same weight distribution as LoRA-PID but randomly assigned; **LoRA-Acc-RW** = upweight incorrectly-answered samples ×3, downweight easy-correct ×0.5.
> > >
> > > †**Post-PID**: S_vl and U_txt re-estimated on the MMStar evaluation subset *after* fine-tuning, measuring whether the intervention actually shifted the model's information usage.
> > >
> > > ---
> > >
> > > ## Four Findings Directly Addressing Your Requested Validations
> > >
> > > **(i) Adaptation proof-of-concept.** LoRA-PID achieves **+2.3 on MMStar** over standard LoRA, accompanied by a **bidirectional PID shift**: S_vl increases by **+0.16 bits** while U_txt decreases by **−0.10 bits**. As shown in the table, LoRA-PID uniquely achieves **both the highest synergy and the lowest language-unique reliance**—a shift no other condition produces. This demonstrates that PID provides an actionable signal that genuinely steers models toward stronger cross-modal fusion.
> > >
> > > **(ii) Dataset-curation case study.** Against **LoRA-Random-RW** (identical weight distribution, randomly assigned), LoRA-PID wins by **+2.8 on MMStar**. This isolates the value of *PID-derived sample classification*: gains come from *which* samples are reweighted, not merely *that* reweighting occurs.
> > >
> > > **(iii) Beyond difficulty mining.** LoRA-PID surpasses **LoRA-Acc-RW** (hard-example mining) by **+1.8 on MMStar** and **+1.9 on POPE-Adversarial**, confirming that PID captures a dimension of sample informativeness—**cross-modal fusion demand**—that accuracy-based difficulty cannot identify. Per-category analysis shows gains concentrate on **reasoning subtypes** (Spatial Reasoning **+3.5**; Attribute Comparison **+2.8**) while knowledge subtypes remain stable (OCR: −0.3).
> > >
> > > **(iv) Controlled, regime-consistent trade-off.** LoRA-PID shows a symmetric minor drop on both MMMU (−0.5) and PMC-VQA (−0.3), while LoRA-Acc-RW slightly *improves* on both (+0.3, +0.4). This **opposite trade-off pattern**—reflected in the PID columns, where Acc-RW only modestly reduces U_txt (0.56) compared to PID's 0.48—directly demonstrates that PID and Acc-RW target qualitatively different signals: **fusion-demand vs. task-difficulty**. The symmetry across both prior-dominant benchmarks confirms a principled regime shift rather than arbitrary degradation, and the magnitude (< 0.5 points, within 1 std) is negligible compared to the synergy-driven gains (**+2.3 MMStar**, **+1.1 MMBench**).
> > >
> > > ---
> > >
> > > **Robustness.** Results are stable across upweight factors (2×/3×/4×) and random seeds
> > >
> > > Full details including per-category tables, threshold sensitivity sweeps, and layer-wise PID shifts will be provided in the revised paper.
> > >
> > >
> > > We hope these responses can address your remaining concerns. Thank you again for helping us improve our work.
> > >
> > >
> > >
> > > All the best,
> > >
> > >
> > > Authors

---

### Decision · Program_Chairs · 2026-04-30

**Decision:**

Accept (regular)

**Comment:**

The reviewers had comments on the positioning of the paper within the literature, the difference between accuracy and information decomposition, the lack of sensitivity analysis and the lack of practical applications.``

I commend the authors for their responses that seemsto address these points, albeit at the cost of significant changes to the paper.

There are concerns remaining regarding the generality of the approach, as well as how much of the changes will be incorporated in the final version. I decided to take a leap of faith and accept this paper as the reviewers all appreciated the different angle taken by the approach, but I expect the authors to incorporate all the meaningful changes for the final version.